# No Silver Bullet for Digital Soil Mapping: Country-specific Soil Organic Carbon Estimates across Latin America

Mario Guevara[1], Guillermo Federico Olmedo[2,3], Emma Stell[1], Yusuf Yigini[3], Yameli Aguilar Duarte[4], Carlos Arellano Hernández[5], Gloria E Arévalo[6], Carlos Eduardo Arroyo-Cruz[7], Adriana Bolivar[8], Sally Bunning[9], Nelson Bustamante Cañas[10], Carlos Omar Cruz-Gaistardo[5], Fabian Davila[11], Martin Dell Acqua[11], Arnulfo Encina[12], Hernán Figueredo Tacona[13], Fernando Fontes[11], José Antonio Hernández Herrera[14], Alejandro Roberto Ibelles Navarro[5], Veronica Loayza[15], Alexandra M. Manueles[6], Fernando Mendoza Jara[16], Carolina Olivera[17], Rodrigo Osorio Hermosilla[10], Gonzalo Pereira[11], Pablo Prieto[11], Iván Alexis Ramos[18], Juan Carlos Rey Brina[19], Rafael Rivera[20], Javier Rodríguez-Rodríguez[7], Ronald Roopnarine[21,22], Albán Rosales Ibarra[23], Kenset Amaury Rosales Riveiro[24], Guillermo Andrés Schulz[25], Adrian Spence[26], Gustavo M Vasques[27], Ronald R Vargas[3], and Rodrigo Vargas[1]

[1]University of Delaware, Department of Plant and Soil Sciences, Newark DE, USA. 19713
[2]INTA EEA Mendoza, San Martín 3853, Luján de Cuyo, Mendoza, Argentina, M5507EVY
[3]FAO, Vialle de Terme di Caracalla, Rome, Italy
[4]Instituto Nacional de Investigaciones Forestales, Agrícolas y Pecuarias, Mexico
[5]Instituto Nacional de Estadísitica y Geografía, Aguascalientes, Mexico
[6]Zamorano University of Honduras and Asociación Hondureña de la Ciencia del Suelo
[7]National Commission for the Knowledge and Use of Biodiversity, Mexico City, Mexico
[8]Subdirección Agrología, Instituto Geográfico Agustín Codazzi, Colombia
[9]Oficina Regional de la FAO para América Latina y el Caribe, Chile
[10]Servicio Agrícola y Ganadero, Chile
[11]Direccion General de Recursos Naturales, Ministerio de Ganaderia, Agricultura y Pesca, Uruguay
[12]Facultad de Ciencias Agrarias de la Universidad Nacional de Asunción, Paraguay
[13]Land Viceministry, Ministry of Rural Development and Land, Bolivia
[14]Universidad Autónoma Agraria Antonio Narro Unidad Laguna, Mexico
[15]Ministerio de Agricultura y Ganaderia, Quito, Ecuador
[16]Universidad Nacional Agraria, Nicaragua
[17]Representación de FAO en Colombia
[18]Instituto de Investigación Agropecuaria de Panamá, Panama
[19]Sociedad Venezolana de la Ciencia del Suelo, Venezuela
[20]Ministerio de Medio Ambiente, Republica Dominicana
[21]Department of Natural and Life Sciences, COSTAATT, Port-of Spain, Trinidad and Tobago
[22]University of the West Indies, St Augustine Campus, Trinidad and Tobago
[23]Instituto de Innovación en Transferencia y Tecnología Agropecuaria, Costa Rica
[24]Ministerio de Ambiente y Recursos Naturales de Guatemala
[25]INTA CNIA, Buenos Aires, Argentina
[26]International Centre for Environmental and Nuclear Sciences, University of the West Indies, Jamaica
[27]Embrapa Solos, Rio de Janeiro, Brazil

*Correspondence to:* Rodrigo Vargas (rvargas@udel.edu)

**Abstract.** Country-specific soil organic carbon (SOC) estimates are the baseline for the Global SOC Map of the Global Soil Partnership (GSOCmap-GSP). This endeavor requires harmonizing heterogeneous datasets and building country-specific capacities for digital soil mapping (DSM). We identified country-specific predictors for SOC and tested the performance of five predictive algorithms for mapping SOC across Latin America. The algorithms included: support vector machines (SVM), random forest (RF), kernel weighted nearest neighbors (KK), partial least squares regression (PL), and regression-Kriging based on stepwise multiple linear models (RK). Country-specific training data and SOC predictors (5x5km pixel resolution) were obtained from ISRIC-World-Soil-Information-Institute. Temperature, soil type, vegetation indices and topographic constraints were the best predictors for SOC, but country-specific predictors and their respective weights varied across Latin America. We compared a large diversity of country-specific datasets and models, and were able to explain SOC variability in a range between <1% and <60%, with no universal predictive algorithm among countries. A global (n=11268) ensemble of these algorithms was able to explain $\sim$ 39% of SOC variability from repeated 5 fold cross-validation. We report a combined SOC stock of 77.8 $\pm$43.6 Pg (uncertainty represented by the full conditional response of model-independent residuals). SOC stocks were higher in tropical forests (30 $\pm$16.5 Pg) and croplands (13 $\pm$8.1 Pg). Country-specific and the regional ensemble reveal spatial discrepancies across geopolitical borders, higher elevations and coastal plains, but provide similar stocks (77.8 $\pm$42.2 and 76.8 $\pm$45.1 Pg, respectively). These results are conservative compared to global estimates (e.g., SoilGrids250m 185.8 Pg, the Harmonized World Soil Database 138.4 Pg, or the GSOCmap-GSP 99.7 Pg). Countries with large area (i.e., Brazil, Bolivia, Mexico, Peru) and large spatial SOC heterogeneity had lower SOC stocks per unit area and larger uncertainty in their predictions. We highlight that expert opinion is needed to set boundary prediction limits unrealistic high estimates. . Maximizing explained variance while minimizing prediction bias, selecting predictive algorithms for SOC mapping should consider density of available data and variability of country-specific environmental gradients. This study highlights the large degree of spatial uncertainty in SOC measurements across Latin America. We provide a reproducible framework for improving country-specific mapping efforts and reducing current discrepancy of global, regional and country-specific SOC estimates.

## 1 Introduction

Soils store around 1500 Pg of carbon and represent the largest terrestrial carbon pool (Jackson et al., 2017); thus, it is critical to accurately quantify the variability of soil organic carbon (SOC) from local-to-global scales. During the 4[th] Session of the Global Soil Partnership (GSP) Plenary Assembly held in May 2016 in Rome, it was agreed to develop a Global Soil Organic Carbon Map (GSOCmap) (FAO, 2017). The overarching goal is that a Global SOC Map of the Global Soil Partnership (GSOCmap-GSP) will be developed using a distributed approach relying on country-specific SOC maps. Country-specific maps represent a valuable source of information to explain the high discrepancy of current global SOC estimates (e.g., the SoilGrids250m system and the Harmonized World Soil Database, see (Tifafi et al., 2018)). The Food and Agriculture Organization (FAO) recently compiled how different statistical methods (e.g., regression-kriging and machine learning) could be used to generate country-specific SOC maps and calculate uncertainty (Yigini et al., 2018). All these approaches consider the reference framework of the SCORPAN model for digital soil mapping (DSM; McBratney et al. (2003)). In the SCORPAN reference framework a

soil attribute (e.g., SOC) can be predicted as a function of the soil forming environment, in correspondence with soil forming factors from the Dokuchaev hypothesis and Jenny's soil forming equation based on climate, organisms, relief, parent material and elapsed time of soil formation (Florinsky, 2012). The SCORPAN (Soils, Climate, Organisms, Parent material, Age and (N) space or spatial position, see (McBratney et al., 2003)) reference framework is an empirical approach that can be expressed as in Eq. (1):

$$Sa_{[x;y\ t]} = f(S_{[x;y\ t]}, C_{[x;y\ t]}, O_{[x;y\ t]}, R_{[x;y\ t]}, P_{[x;y\ t]}, A_{[x;y\ t]}) \tag{1}$$

where $Sa$ is the soil attribute of interest at a specific location N (represented by the spatial coordinates of field observations $x$; $y$) and representing a specific time frame ($t$); $S$ is the soil or other soil properties that are correlated with $Sa$; $C$ is the climate or climatic properties of the environment; $O$ are the organisms, vegetation, fauna or human activity; $R$ is topography or landscape attributes; $P$ is parent material or lithology; and $A$ is the substrate age or the time factor. To generate predictions of $Sa$ across places where no soil data is available, $N$ should be explicit for the information layers representing the soil forming factors. These predictions will be representative of the time period ($t$) when soil available data was collected. Therefore, the prediction factors ideally should represent, the conditions of the soil forming environment for the same period of time (as much as possible) when soil available data was collected. In Eq. (1) the left side is usually represented by the available geo-spatial soil observational data (e.g., from legacy soil profile collections) and the right side of the equation is represented by the soil prediction factors. These prediction factors are normally derived from four main sources of information: a) thematic maps (i.e., soil type, rock type, land use type); b) remote sensing (i.e., active and passive); c) climate surfaces and meteorological data; and d) digital terrain analysis or geomorphometry. The SCORPAN reference framework is widely used, but one critical challenge is to quantify the relative importance of the soil forming factors (i.e., prediction factors) that could explain the underlying soil processes controlling the spatial variability of a specific soil attribute (i.e., SOC).

Arguably, there are two visions for statistical modeling (Breiman, 2001) that influence the predictions of the spatial variability of SOC. One assumes that the variability of observations can be reproduced by a given stochastic data model (e.g., with hypothesis about the spatial structure of the variable). The other uses algorithms and treats as unknown the mechanisms generating the structure of values in available datasets (e.g., with hypothesis about the statistical distribution and moments of the variable). For SOC modeling, the accuracies of global models compared with country-specific estimates have not been evaluated on detail. While globally available SOC predictions rely on large and complex multivariate spaces to represent the soil forming environment, local (i.e., more simple models) may be useful for validation purposes and required to measure the bias of global SOC estimates at particular sites/countries where SOC drivers may be easier to identify due to a smaller range of SOC variance. In addition, the assumptions of global models compared with local efforts may be different, and the quality of local datasets may be higher that sources for global information. Different mapping approaches use a set of given available predictors in different ways. Thus, comparing different approaches and methods is useful to quantify the relative importance of prediction factors across data configurations and distributional properties. We argue that a systematic analysis of predictive

algorithms and consequently selection of predictors (by each one of the algorithms) could provide insights about the underlying factors that control the spatial variability of SOC.

The last decade has seen an increasing diversity of approaches for DSM. Data mining techniques have been successfully used to model and predict the spatial variability of soil properties (Rossel and Behrens, 2010; Hengl et al., 2017; Shangguan et al., 2017) and generate some country-specific SOC maps (Viscarra Rossel et al., 2014; Adhikari et al., 2014). The combination of regression modeling approaches with geostatistics of model residuals (i.e., regression Kriging) is a combined strategy that has been widely used to map SOC (Hengl et al., 2004; Mishra et al., 2009; Marchetti et al., 2012; Kumar et al., 2012; Peng et al., 2013; Adhikari et al., 2014; Yigini and Panagos, 2016; Nussbaum et al., 2014; Mondal et al., 2017). Machine learning algorithms such as random forests or support vector machines have also been used to increase statistical accuracy of soil carbon models (Martin et al., 2011; Hashimoto et al., 2017; Hengl et al., 2017) including applications for SOC mapping (Grimm et al., 2008; Sreenivas et al., 2016; Yang et al., 2016; Hengl et al., 2017; Delgado-Baquerizo et al., 2017; Ließ et al., 2016; Viscarra Rossel et al., 2014). Machine learning methods do not necessarily allow to extract information about the main effects of prediction factors in the response variable (e.g., SOC); consequently, a variable selection strategy is always useful to increase the interpretability of machine learning algorithms. With this diversity of approaches one constant question is if there is a method that systematically improves the prediction capacity of the others aiming to predict SOC across large geographic areas (e.g., Latin America). We postulate that probably there is no universal method (i.e., silver bullet) for DSM, but both global and country-specific efforts are needed to test a variety of predictive algorithms including variable and parameter selection strategies for maximizing explained variance while minimizing prediction bias.

Across Latin America, site or region-specific modeling efforts report high explained variance mapping SOC (Reyes-Rojas et al., 2018). SOC maps are required to quantify SOC stocks and identify areas with the potential for soil carbon sequestration, and distinguish them from areas with high SOC. However, site specific efforts to map SOC across the Argentinean Pampas highlight the challenge of predicting pedologically sound soil maps due to the complexity of SOC spatial variability (Angelini et al., 2016), including the inconsistencies of using simple linear approaches to explain soil and depth interrelationships (Angelini et al., 2017). Site-specific SOC mapping efforts across Brazil suggest that variable selection and the spatial detail of SOC prediction factors are also contributing with the inconsistencies of SOC prediction accuracy (Samuel-Rosa et al., 2015). The constant challenge is how to increase SOC prediction accuracy while also reducing the granularity of SOC grids. The use of high performance computing through open source platforms (i.e., Google Earth) represent a valuable resource to make and continuously update (as new and better data become available) fine grained SOC predictions across countries (Padarian et al., 2017). These SOC predictions are required to build baseline reference estimates to quantify SOC stocks and contribute with better parameterization for projections of SOC under future weather and land degradation scenarios. Therefore, SOC estimates should be ideally based on all available information for each country or region of interest, from both national and global information sources. However, the availability of public SOC information is limited across large areas of Latin America and large discrepancy exist in current global SOC estimates (Tifafi et al., 2018). Thus, there is a pressing need to validate the accuracy of global SOC estimates and contribute with the capacity of countries to meet the GlobalSoilMap specifications (Arrouays et al., 2017) to inform policy decisions around climate change mitigation strategies.

The overarching goal of this study is to compare different predictive algorithms across 19 data/country scenarios with publicly available information to support the development of country-specific SOC maps to be included in the GSOCmap-GSP. Currently, SOC information across Latin America is derived from global models such as the SoilGrids system, or the Harmonized World Soil Database (Hengl et al., 2017; Köchy et al., 2015), which lack quantification of uncertainty and large areas remain parameterized with limited country-specific information. This challenge is not unique for Latin America as many regions around the world (e.g., Africa, Siberia) have limited SOC information to parameterize models to simulate the soil carbon pool. To inform future SOC mapping efforts, this study addresses two specific questions: a) Which environmental variables (derived from publicly available information) have the highest correlations with country-specific SOC information?; and b) Which is the best method (i.e., predictive algorithm) to represent SOC across Latin America and within each country? We assume that methods should inform each other as they are able to explain different aspects of SOC variability. The ultimate aim of this study is to empower capacities for digital SOC mapping across Latin America, and to contribute with the discussion about the importance of integrating country-specific information for representing and predicting soil-related variables (e.g., SOC) to improve regional to global SOC predictions.

## 2 Methods

We base our methodological approach in public sources of information and transparent methods implemented on open sources platforms for statistical computing. Thus, our statistical framework for modeling SOC stocks (illustrated in Fig. 1) could be reproduced across the world for comparative purposes between country specific and global estimates.

### 2.1 SOC observations

Soil organic carbon information was extracted from the WoSIS soil profile database. This dataset represent a great harmonization effort in which a large number of national legacy datasets have been brought together. It includes local-to-national soil profile collections with a sampling strategy generally based on morphological soil attributes (Batjes et al., 2017). The goal of the GSOCmap-GSP is to produce global information for the first 30 cm; thus, we generated synthetic horizons for this depth using a mass preserving spline approach (Bishop et al., 1999). We applied a pedotransfer function based on organic matter (OM) if the bulk density (BLD) information was missing: BLD = 1/(0.6268 + 0.0361 * OM) (Yigini et al., 2018). We decided to use this equation because showed less extreme values than other available pedotransfer functions during preliminary discussion and training exercises (data not shown). Another reason is that there is not a single pedotransfer function applicable to all conditions in Latin America. This equation is representative for soils with organic matter content between 0.17 to 13.5% (Drew, 1973). For coarse fragments (CRFVOL), a value of 0% was used for missing information prior to the mass preservative spline modeling.SOC estimates (0 to 30 cm) were derived following a standardized SOC calculation method (Nelson and Sommers, 1982) (Eq. 2):

$$SOC_{stock} = \frac{ORC}{1000} \times \frac{H}{100} \times BLD \times \frac{(100 - CRFVOL)}{100} \qquad (2)$$

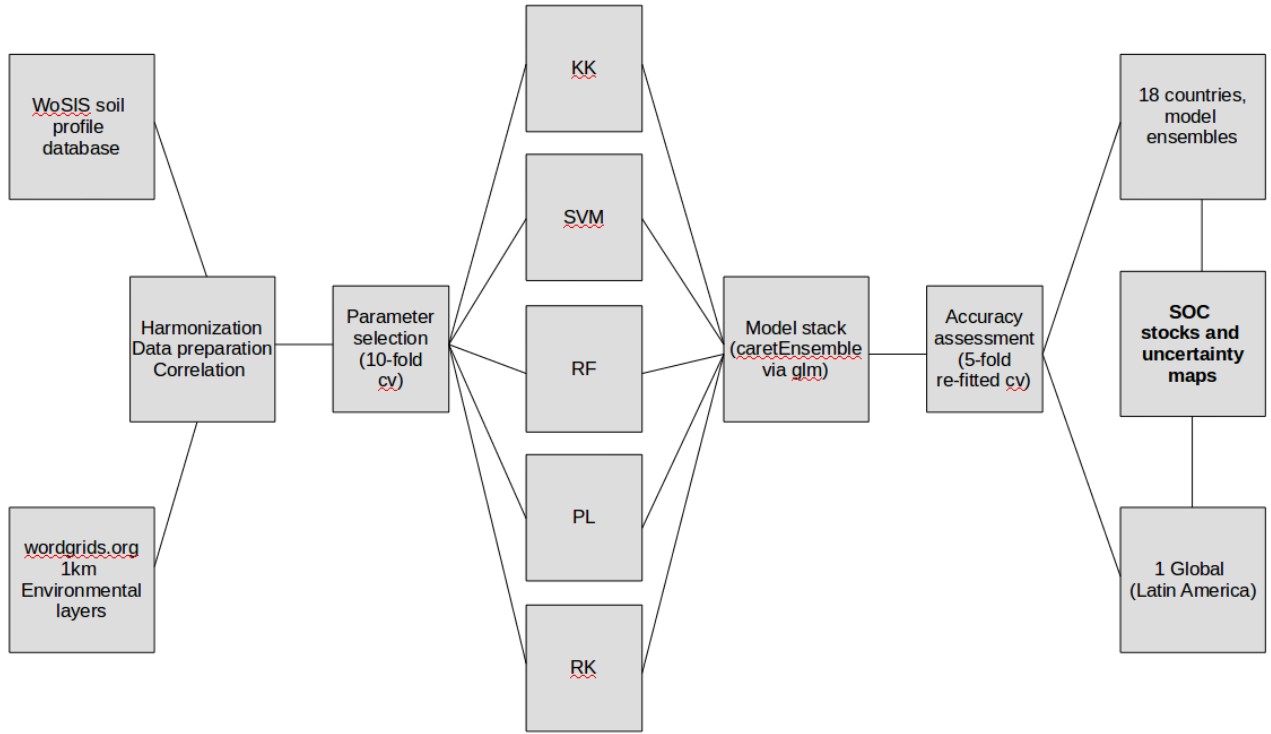

**Figure 1.** Flow diagram of the main methodological steps that we performed in order to generate the country-specific and global SOC predictions. The WoSIS dataset was harmonized with the worldgrids.orgenvironmental data using 5km grids. SOC stocks were calculated at points and correlated predictors identified. Five methods were parameterized and ensembled using a generalized linear model. Accuracy of models and ensembles was assessed with repeated cross validation and country-specific and global (Latin America) ensembles were compared. KK kernel weighted nearest neighbors, SVM support vector machines, RF random forests, PL partial least squares regression, RK regression kriging.

where $ORC$ is SOC density $(\mathrm{g \cdot kg^{-1}})$ and $H$ is soil depth $(30\,\mathrm{cm})$.

Because the limitations and uncertainty in the available BD and CRFVOL data, we also include an error approximation of SOC estimates. This error was derived using Global Soil Information Facilities (GSIF, (Hengl, 2017)) as explained in the next section.

## 2.2 SOC error estimates

The GSIF approach to estimate SOC (function OCSKGM) includes an approximate error which we use to quantify the reliability of SOC estimates (Hengl et al., 2017). This error is approximated using the Taylor Series Method, by a truncated Taylor series centered by the means as explained in previous studies (Heuvelink, 2018). We map the error trend of SOC estimates by interpolating the values in a country basis using the generic framework for predictive modeling based on machine learning and

buffer (geographical) distances (Hengl et al., 2018). We followed this method to provide a spatial explicit measure of the SOC estimation error. We use this method because it can be implemented without prediction factors (e.g., only buffer distances) and because it is practically free of assumptions but consider the geographical proximity to and composition of the sampling location points as explained by its developers (Hengl et al., 2018). SOC error estimates represent a component of uncertainty1

the overall quality of country specific input data.

## 2.3   SOC training data and exploratory analysis

Each country-specific SOC dataset was transformed to its natural logarithm to reduce the right-skewed distribution of SOC values and because exploratory analysis showed that this transformation can improve the prediction capacity of further modeling methods. To analyze the statistical distribution of SOC values, a probability distribution function was plotted and a

Shapiro-Wilk test of normality was conducted on each dataset. The units of the SOC estimates are $\mathrm{kg \cdot m^{-2}}$. Our global (Latin America) dataset of 11268 SOC estimates was divided using a simple bootstrapping technique (Kuhn. et al., 2017) and 25% of data was used for independent validation purposes, and the remaining 75% of data for training prediction models. We couple this information with a public source of prediction factors.1

## 2.4   Soils prediction factors

We used environmental information from WorldGrids (worldgrids.com), which is an initiative of ISRIC-World Soil Information. We downloaded and masked 118 environmental layers (i.e., prediction factors) for each country to quantitatively represent the soil forming environment. The prediction factors were harmonized into a 1x1km global grid by the WorldGrids project from three main information sources: remote sensing, climate surfaces, and digital terrain analysis (http://worldgrids.org/doku.php/wiki:layers). Additional terrain parameters (e.g., terrain slope, aspect, catchment area, channel network base level, terrain curvature, topo-

graphic wetness index, length-slope factor) from elevation data were calculated in SAGA GIS for each country following the standard implementation for basic terrain parameters (Conrad et al., 2015). We re-sampled the prediction factors into a 5x5km pixel size grid to reduce the computational demand required to make predictions and facilitate the reproducibility of this DSM framework without the need of High Performance Computing to mke predictions of SOC.

## 2.5   Prediction of SOC

We predict in a country specific and in a regional (Latin American) basis. We base our prediction framework in the following six steps:

   – First, the relationship between SOC and prediction factors was explored using simple correlation analysis.

   – Second, the 10 prediction factors with highest correlations with SOC data were identified for each country and used for further analyses.

– Third, we explore, parameterize and compare five statistical methods with different assumptions to model SOC variability across Latin America: Regression-Kriging (based on a multiple linear regression model (RK) and partial least squares

regression (PLS), support vector machines (SVM), random forests (RF), and kernel weighted nearest neighbors (KK). A brief explanation for each modeling approach is provided in Appendix A.

- Fourth, we re-fit the aforementioned models and using the caretEnsemble tools for stacking models (Deane-Mayer and Knowles, 2016; Kuhn. et al., 2017). The caretEnsemble approach uses the RMSE to weight and ensemble regression models under a generalized linear model to create a linear blend of predictions. The RMSE was derived from the residuals of the new models via repeated 5-fold cross validation.

- Fifth, we calculate independent model residuals (by predicting to the 25% not used for modeling). For each 5x5km pixel we estimate the full conditional response of these residuals to the SOC prediction factors following the quantile regression method available within the quantregForest modeling framework (Meinshausen, 2017, 2006). We use this map as surrogate of model uncertainty complementary to the approximated error trend of SOC estimates.

- Sixth, we use all Latin American data in the WoSIS system to repeat the fourth and fifth steps of our modeling framework, generating regional predictions of SOC and comparing with country-specific results and global estimates. We also evaluate the prediction capacity of these models.

## 2.6 Model evaluation and accuracy

First, for each single model we perform a 10-fold validation strategy following a generic recommendation (Borra and Di Ciaccio, 2010) to select the optimal model parameters. For each model the train function of the caret package (Kuhn. et al., 2017) includes simple re-sampling techniques for automatic model parameter selection (see parameter description Appendix A). Thus we obtained unbiased residuals for each model on each country that we compared using Taylor diagrams (Carslaw and Ropkins, 2012). A Taylor Diagram summarizes multiple aspects of model performance, such as the agreement and variance between observed and predicted values (Taylor, 2001). In a Taylor Diagram, each model is represented by a point in the plot describing how well the patterns of observed and modeled match each other. Two models will have a similar predictive capacity if they overlap across the intersection of an error vector, a variance ratio and a correlation vector.

We analyzed the overall ratio ($EC_r$) between model errors (RMSE) and the correlation between observed and predicted values (corr) for each model across all countries. We propose this ratio $EC_r$ as an approach to better understand the agreement between the correlation (calculated by the means of cross validation) and the RMSE (derived from the unbiased residuals of cross validation). Before calculating the RMSE/correlation ratio, the RMSE and the correlation between observed and predicted were standardized (by its maximum and minimum values) to a range between 0 and 1 using:

$$\text{RMSE}_{std} = \frac{\text{RMSE}_i - min(\text{RMSE})}{range(\text{RMSE})} \tag{3}$$

$$\text{corr}_{std} = \frac{\text{corr}_i - min(\text{corr})}{range(\text{corr})} \qquad (4)$$

$$EC_r = \frac{\text{RMSE}_{std}}{\text{corr}_{std}} \qquad (5)$$

Where $EC_r$ is the proposed ratio between errors and correlation between observed and predicted; $\text{RMSE}_i$ is the observed RMSE for the $i$th model; $min(\text{RMSE})$ is the minimum observed value of RMSE, and $range(\text{RMSE})$ is the difference between the maximum and minimum observed values of RMSE; $\text{corr}_i$ is the observed correlation for the $i$th model; $min(\text{corr})$ is the minimum observed value of correlation, and $range(\text{corr})$ is the difference between the maximum and minimum observed values of correlation

If the value of the $EC_r$ was close to 0, then there is a stronger agreement between high RMSE and low correlation, or low RMSE and high correlation. If this value deviated from 0 (up to 1 or more), then the RMSE would tend to be high while the correlation was also high, suggesting that the method represents the variability of SOC but with high bias.

Model accuracy (also represented by the RMSE and $R^2$) was assessed for the model ensembles with a more strict (but computationally expensive) 5-fold and five times repeated cross validation strategy. This model re-fitting allows more stable accuracy results with the ultimate goal of comparing country-specific and global (Latin America) estimates. Repeated 10 and 5-fold cross validation have both been used to compare both machine learning and geo-statistical approaches for mapping soil properties from book examples to real applications at the global scale (Hengl et al., 2018, 2017). In addition, model independent residuals were obtained also from the 25% of data not used in the coutry-specific and global ensembles to estimate a spatial explicit measure of uncertainty (as explained in the step five of our prediction framework).

## 2.7 SOC stocks

First we analyzed the influence of the maximum allowed prediction limits for each prediction algorithm. The sensitivity of the total SOC stock to the model prediction limit was tested by increasing (every 10) the maximum prediction limit from $0.5 \, \text{Ton} \cdot \text{Ha}$ until finding a stable rate. Geopolitical limits were obtained from the Global Administrative areas project (https://gadm.org/). Using these country limits we report our country-specific and Latin American SOC estimates. For comparative purposes we also extract for each country the global SOC estimates from the SoilGrids system (Hengl et al., 2017), the Harmonized World Soil Database (Köchy et al., 2015) and the GSOCmap-GSP (see http://54.229.242.119/apps/GSOCmap.html). We also report stocks across the land cover classes derived from the Latin American Network for Monitoring and Studying of Natural Resources, (a product with an estimated accuracy of 84% (Blanco et al., 2013)). We report the overall uncertainty of these stocks as the sum of the model independent residual map and the approximated error trend of the SOC estimates. Some no data countries were filled with the average of surrounding extent SOC predictions. All analyzes were performed using the R language for statistical computing (R Core Team, 2017).

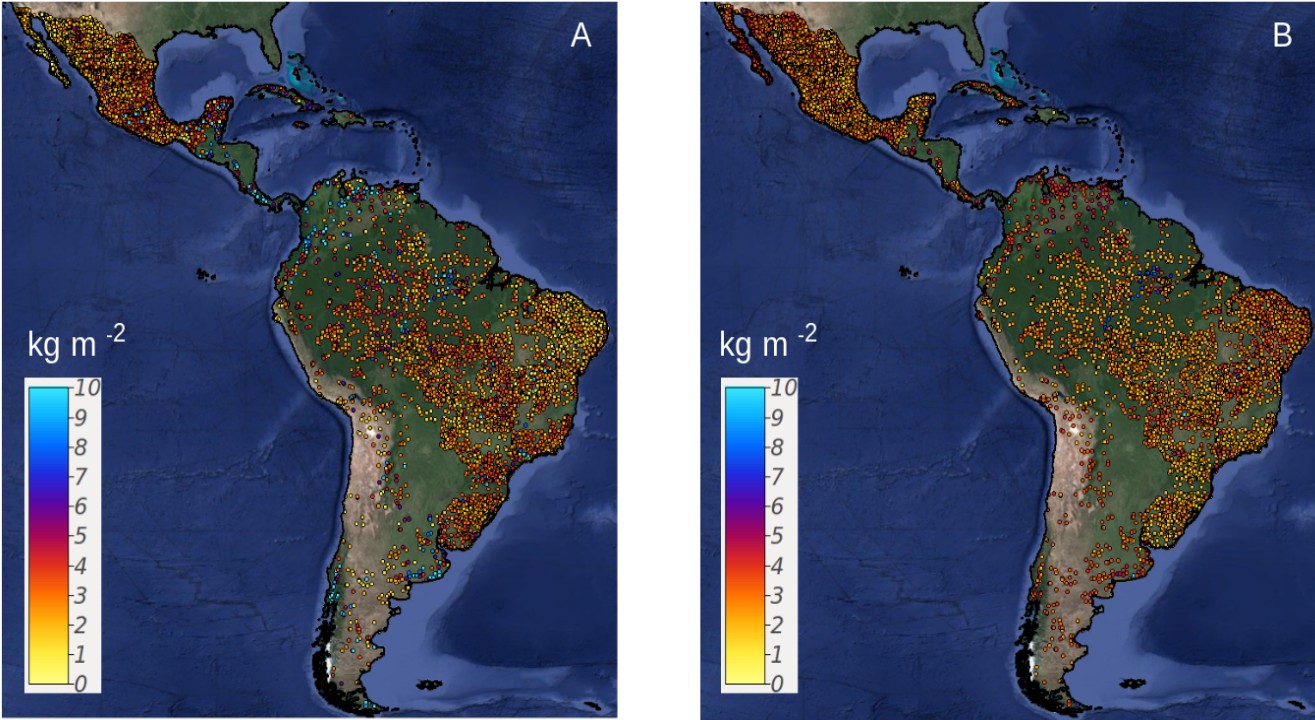

**Figure 2.** Spatial distribution of available SOC in WoSIS for Latin America, in A the SOC estimates and in B the approximated error based on Taylor series as implemented in the R-GSIF package.

## 3 Results

### 3.1 Descriptive statistics

SOC across different countries showed a wide diversity of data-scenarios (Table 1). Costa Rica (with a mean of $11.05 \ \mathrm{kg \cdot m^2}$), Chile (with a mean of $9.88 \ \mathrm{kg \cdot m^2}$) and Colombia (with a mean of $8.15 \ \mathrm{kg \cdot m^2}$) are the countries with the highest SOC values. Brazil (n=5616) and Mexico (n=4321) were the countries with highest data availability. In contrast, Honduras (n=11), Guatemala (n=20) and Belize (n=21) were the countries with less density of of SOC estimated values (Table 1). With the original (untransformed) dataset, the only countries that showed a normal distribution after the Shapiro- Wilk test of normality with an alpha of 0.05 were Belize, Guatemala, Honduras and Suriname.

### 3.2 Spatial distribution and point error estimates

There are large areas of Latin America with no available SOC observational data in the WoSIS system (e.g., Chile, Argentina and Central America). We found significant error estimates across large areas with high density of SOC data but low carbon contents, such as northern Mexico or the Brazilian semiarid savanna located at the eastern side of the country (Fig. 2)..

**Table 1.** Descriptive statistics of SOC estimates $kg \cdot m^2$ and total land area for each analyzed country. N is the number of observations. We provide quantiles, median, mean and the standard deviation of SOC data. The columns p and plog represent the probability values derived from the Shapiro-Wilk test of normality before (p) and after (plog) the log transformation of SOC values. When p is larger than plog, the log transformation of the data did not increased the probability of normality in the dataset. For comparative purposes we provide (as a supplementary Figure S1) the probability distribution functions of available data before and after the log transformations. ARG=Argentina, BLZ=Belize, BOL=Bolivia, BRA=Brazil, CHL=Chile, COL=Colombia, CRI=Costa Rica, CUB=Cuba, ECU=Ecuador, ESP=Espana, GTM=Guatemala, HND=Honduras, JAM=Jamaica, MEX=México, NIC=Nicaragua, PAN=Panama, PER=Peru, SUR=Suriname, SLV=El Salvador, URY=Uruguay, VEN=Venezuela.

| Country | n | Land Area (km2) | Min | 1st Q | Med | Mean | 3rd Q | Max | SDev | p / plog |
|---|---|---|---|---|---|---|---|---|---|---|
| ARG | 231 | 2736690 | 0.34 | 1.88 | 3.21 | 5.65 | 5.96 | 86.85 | 9.33 | <0.001 / 0.03 |
| BLZ | 21 | 22970 | 1.84 | 4.49 | 6.72 | 7.71 | 9.99 | 19.48 | 4.32 | 0.08 / 0.99 |
| BOL | 76 | 1083301 | 0.64 | 1.83 | 2.56 | 2.64 | 3.20 | 7.65 | 1.21 | <0.001 / 0.08 |
| BRA | 5616 | 8358140 | 0.07 | 1.99 | 2.67 | 3.23 | 3.34 | 573.76 | 9.18 | <0.001 / <0.001 |
| CHL | 44 | 743812 | 0.43 | 3.58 | 5.19 | 9.88 | 16.52 | 31.87 | 8.86 | <0.001 / 0.01 |
| COL | 166 | 1038700 | 0.66 | 3.44 | 5.78 | 8.15 | 9.95 | 52.62 | 7.35 | <0.001 / 0.96 |
| CRI | 43 | 51060 | 2.27 | 4.07 | 7.23 | 11.05 | 10.85 | 82.57 | 14.90 | <0.001 / 0.001 |
| CUB | 48 | 109820 | 0.36 | 2.85 | 3.61 | 4.32 | 5.73 | 10.98 | 2.23 | 0.004 / <0.001 |
| ECU | 77 | 276841 | 0.99 | 2.37 | 3.65 | 5.15 | 4.36 | 24.36 | 5.15 | <0.001 / <0.001 |
| GTM | 20 | 107159 | 2.60 | 5.66 | 8.48 | 7.73 | 9.75 | 12.41 | 3.11 | 0.14 / 0.007 |
| HND | 11 | 111890 | 2.69 | 5.25 | 6.48 | 6.71 | 8.32 | 12.38 | 2.78 | 0.72 / 0.39 |
| JAM | 76 | 10831 | 1.29 | 3.01 | 3.99 | 4.35 | 4.83 | 12.90 | 1.99 | <0.001 / 0.72 |
| MEX | 4321 | 1943945 | 0.00 | 1.73 | 2.49 | 2.56 | 3.25 | 35.55 | 1.49 | <0.001 / <0.001 |
| NIC | 26 | 119990 | 2.93 | 3.94 | 7.31 | 7.50 | 9.04 | 15.91 | 3.78 | 0.05/0.09 |
| PAN | 25 | 74177 | 3.39 | 4.90 | 7.53 | 7.59 | 9.13 | 19.89 | 3.76 | 0.003 / 0.49 |
| PER | 145 | 1279996 | 0.19 | 1.89 | 2.93 | 2.92 | 3.55 | 8.35 | 1.42 | 0.005 / <0.001 |
| SUR | 27 | 156000 | 1.38 | 2.60 | 3.35 | 3.37 | 4.07 | 6.01 | 1.20 | 0.69 / 0.51 |
| URY | 130 | 175015 | 0.82 | 2.70 | 3.38 | 4.34 | 3.90 | 46.54 | 4.67 | <0.001 / <0.001 |
| VEN | 164 | 882050 | 0.31 | 2.58 | 4.14 | 5.92 | 6.57 | 44.35 | 6.37 | <0.001 / 0.11 |

## 3.3 Correlation of SOC and its predictors

Best correlated predictors were not the same across countries. We found higher correlations with the original data sets transformed to its natural logarithm, as data had a right-skewed distribution and did not follow a normal distribution (i.e., log-normal). Highest correlations of available SOC data and its environmental predictors were associated with temperature-related-variables across Honduras, Costa Rica, Peru, Chile, Guatemala and Suriname (the $r^2$ varied from from 0.35 to 0.58). However, there were a low number of available SOC observations across these countries in the WoSIS system (between 11 to 34).

Similarly, across countries with high data availability (e.g., Mexico and Brazil) the strongest correlations between SOC and prediction factors were associated with temperature-related variables (Table 2). In all cases, the relationship between SOC and temperature-related variables was negative. In contrast, SOC had a positive relationship with elevation-derived terrain parameters ($r^2$ varied from 0.43 to 0.59) such as terrain curvature, potential incoming solar radiation, and slope of terrain.

Lower correlations of SOC data with prediction factors were found across Brazil, Bolivia, Uruguay, Cuba, Panama, Venezuela and Argentina (e.g.$r^2$ <0.2). The correlation analysis was useful to formulate a working hypothesis about the major drivers of the spatial variability of SOC across countries based on our DSM conceptual framework (e.g. $SOC_{ARG} = f$ [px4wcl3a + px3wcl3a + evmmod3a + l07igb3a + px2wcl3a + ...]). For example, the best correlated predictors with SOC for Argentina were precipitation-related variables (px4wcl3a, px3wcl3a, px2wcl3a), remote sensing based vegetation indexes (evmmod3a), and a

probability-based shrubland map (l07igb3a) (Table 2) (see sources of this maps in http://worldgrids.org/doku.php/wiki:layers).

### 3.4 SOC related properties

Correlations between ORCDR and prediction factors were higher with maximum and mean night-time temperature, where Costa Rica and Chile had the highest correlations ($r^2$ varied from 0.61 to 0.71). The best correlated variables with BLD were terrain parameters: relative slope position, vertical distance to channel network, flow accumulation areas, and potential

incoming solar radiation. These correlations were stronger across Guatemala, Belize and Panama ($r^2$ varied from 0.52 to 0.67). We found that terrain slope and the standard deviation of temperature were the variables with highest correlations with CRFVOL; where Nicaragua, Honduras and Argentina had the highest correlations ($r^2$ varied from 0.40 to 0.55). We did not found a dominant algorithm to predict SOC related properties. Slightly higher correlations between observed and predicted values were achieved with RF, but in most cases different methods showed similar prediction capacity. The highest prediction

error was found with RK for CRFVOL, but for all other output variables all prediction algorithms had a similar range of errors (Fig. 3). The PLS and SVM had the lowest variance for prediction of each one of the four soil properties. The $r^2$ values for predicting the combined SOC related properties (ORCDR, CRFVOL and BLD) for each prediction algorithm where: RK($r^2$ 0.67 to 0.76), RF($r^2$ 0.56 to 0.74), SVM ($r^2$ 0.32 to 0.71), PL ($r^2$ 0.46 to 0.69) and KK ($r^2$ 0.19 to 0.64). Across countries with lower data availability and sparse distribution SVM and RK algorithms resulting in lower model performance.

### 3.5 Country-specific SOC predictions

We did not find a dominant algorithm to predict SOC in a country-specific basis (Fig. 4). Overall, machine learning prediction algorithms generated similar results. Higher agreement of machine learning prediction algorithms was found in small countries where environmental conditions and land cover/use characteristics tend to be more homogeneous (e.g. Jamaica, Suriname). RK showed higher discrepancies in countries where data distribution was sparse (e.g., Suriname, Chile, Guatemala), but was

effective across countries with higher and/or well distributed data availability (e.g., Mexico, Brazil). Machine learning SOC predictions were conservative compared with RK (RK generated the higher density of extreme and unreliable SOC values). PL had comparable results with machine learning algorithms (i.e., KK, SVM, RF). From the cross-validation strategy, higher $r^2$ values between observed and predicted data were found for Costa Rica (0.58; n=21) using SVM while the lowest error was

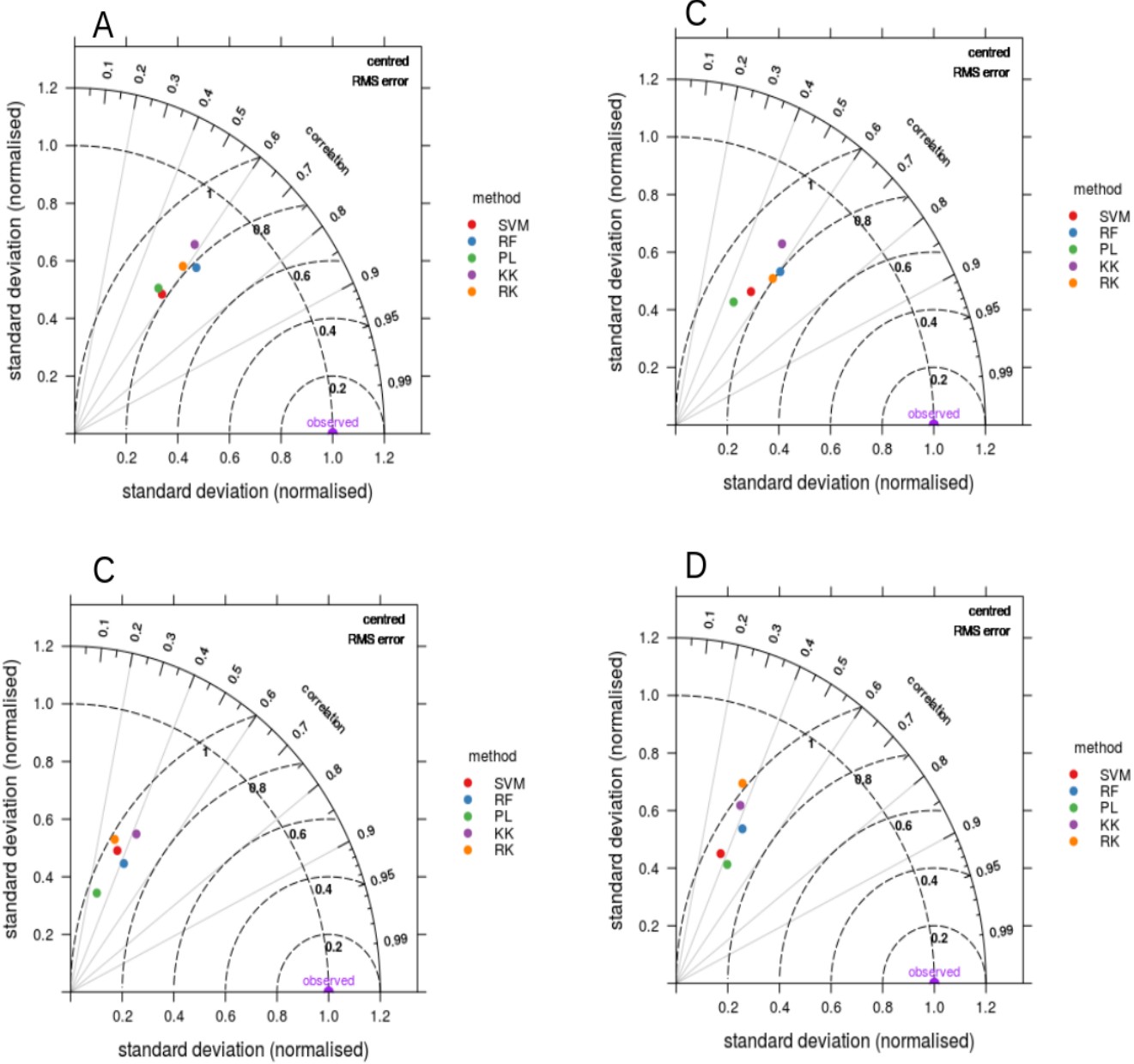

**Figure 3.** Taylor diagram showing the performance of the 5 models evaluated. In A SOC stock, in B SOC density (ORCD), in C bulk density (BLD) and in D coarse fragments (CRFVOL) (OCR, BLD and CRFVOL), This analysis is based in all available data across Latin America. Note that although RF tend to generate higher correlation, it also shows high variance in predictions. Note how the points are close each other and that the differences on accuracy between them generally falls within the same intersection of error, variance and correlation, suggesting a similar prediction capacity of the implemented approaches.

found Suriname ($0.36 \, \mathrm{kg \cdot m^{-2}}$; n =37) using PL. In contrast, algorithms had lower prediction capacity for countries with large areas (e.g., Brazil, Mexico) despite the large data availability.

The simple correlation (main effect) between the $r^2$ and rmse for RF, PL, KK and RK was positive (0.18, 0.35, 0.32 0.1; respectively). In contrast, this correlation was stronger for SVM (but negative; -0.65) where increasing the explained variance resulted in a lower error. These results suggest a low level of agreement between these two information criteria ($r^2$ and rmse) commonly used on DSM to assess performance of prediction algorithms.

Agreement between the rmse and $r^2$ was found only in 12 of the 19 countries, resulting in country-specific "recommended" prediction algorithms. Here we list the prediction algorithms that generated the best correlation and the best rmse for each country: ARG (RK, RK), BLZ (RF, RK), BOL (SVM, KK), BRA (RF, RF), CHL (PL, PL), COL (RF, RF), CRI (SVM, SVM), CUB (PL, PL), ECU (RK, RK), GTM (KK, RF), HND (SVM, KK), JAM (RF, RF), MEX (RK , RK), NIC (RF, RF), PAN (PL, KK), PER (KK, KK), SUR (SVM, PL), URY (RF, RK) and VEN (RK, RK) (see country codes in Table 1). Brazil and Mexico had the highest number of observations (nearly 80% of the total) and the same method yields the highest $r^2$ and the lowest RMSE. We clarify that the best within country method is not the same for each country. The higher $EC_r$ was found with PL (0.96) followed by RF (0.54) and KK (0.43), informing that these predictive algorithms do not minimize prediction bias while increasing the explained variance. SVM (with 0.008) and RK (with 0.003) had the lowest $EC_r$, informing that they maximize the explained variance while minimizing prediction bias.

## 3.6 Model ensembles and SOC maps

High discrepancy was found among the country-specific SOC predictions and between country-specific and regional SOC predictions. Although both maps predict SOC following the a similar general pattern, the country-specific ensemble shows a higher density of unrealistic patterns across Guatemala, Venezuela, Northern Brazil or the surroundings of Uruguay (Fig. 5A). These areas correspond to areas where we report both higher SOC calculation errors and model uncertainty (Fig. 6).

The regional model shows a smooth pattern and a notable discrepancy predicting higher SOC across higher elevations of Southern Andes and boundaries of the Amazon Basin (Fig. 5B). Based on the 5-fold repeated cross validation, we report a $r^2$ 0.39 value for the regional model and $r^2$ values for the country-specific approach that vary from 0.01 to 0.55.

However, high uncertainty was found across some areas in these countries, mainly in the semiarid regions. Residual uncertainty from independent validation show higher errors across Geopolitical borders (in Chile, Argentina, Colombia, Ecuador and Venezuela and the Brazilian savanna) while the residual uncertainty map from the global model reveal higher uncertainty across ecologically meaningful transitions, with no evident effect of geopolitical borders. The trend of the mean approximated error suggest that higher uncertainty exist in the SOC calculation than in the modeling framework (Fig. 6), however this map is just to visualize the general continuous trend or error estimates based only on buffer distances.

Primarily, the pacific coastal plains, the Amazonas delta, some closed watersheds and wetlands across Mexico and some sparse points across central America showed the higher discrepancies. Mexico and Brazil, with higher density of SOC data, were the countries showing less discrepancy between country and global models (Fig. 8A). The global model showed more conservative across large areas of Latin America, we report that the geographical areas where country-specific models tend to

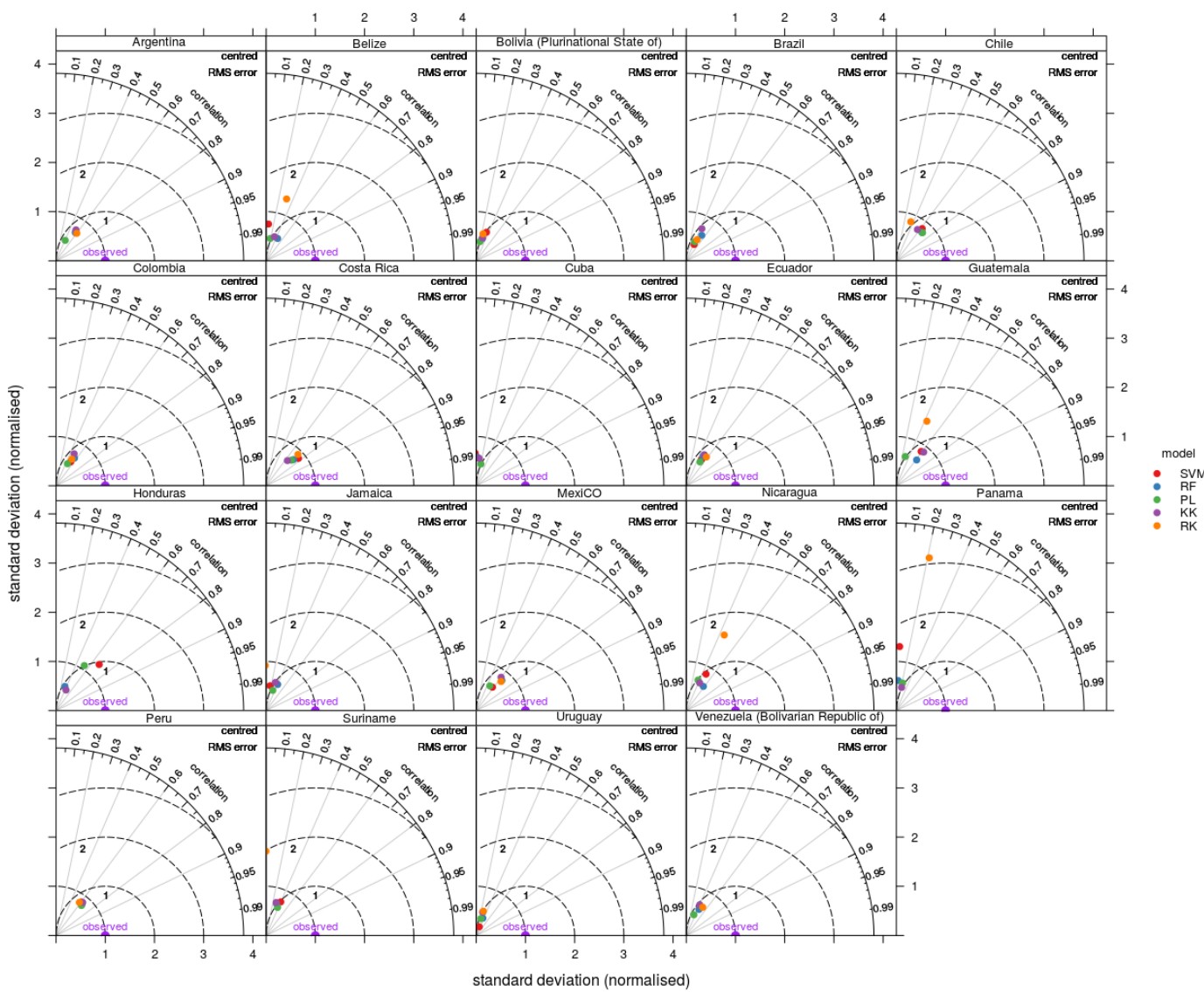

**Figure 4.** Taylor diagram showing the performance of the 5 models evaluated for country-specific SOC estimates across Latin America. Please note that the position of each point/method vary from each dataset to another, suggesting that the predictive capacity changes when data characteristics are different.

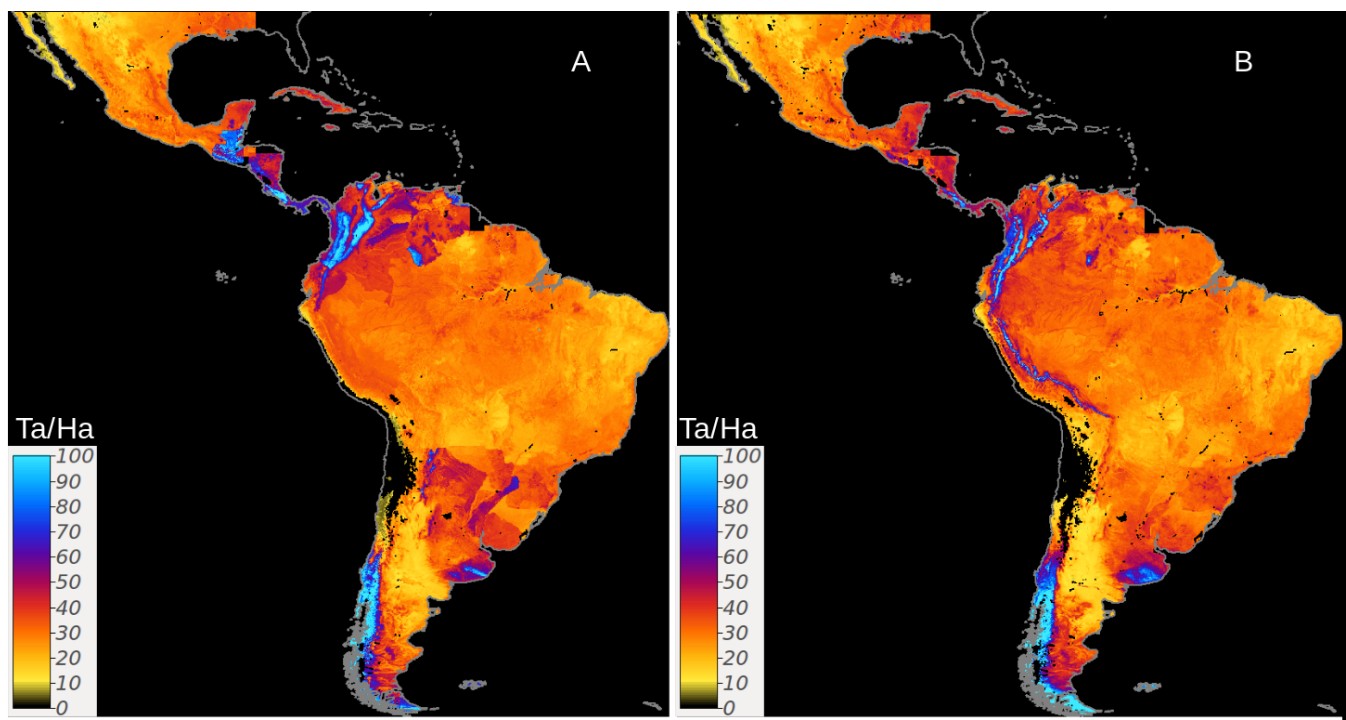

**Figure 5.** A Country-specific ensemble of methods and B global (Latin America) prediction.

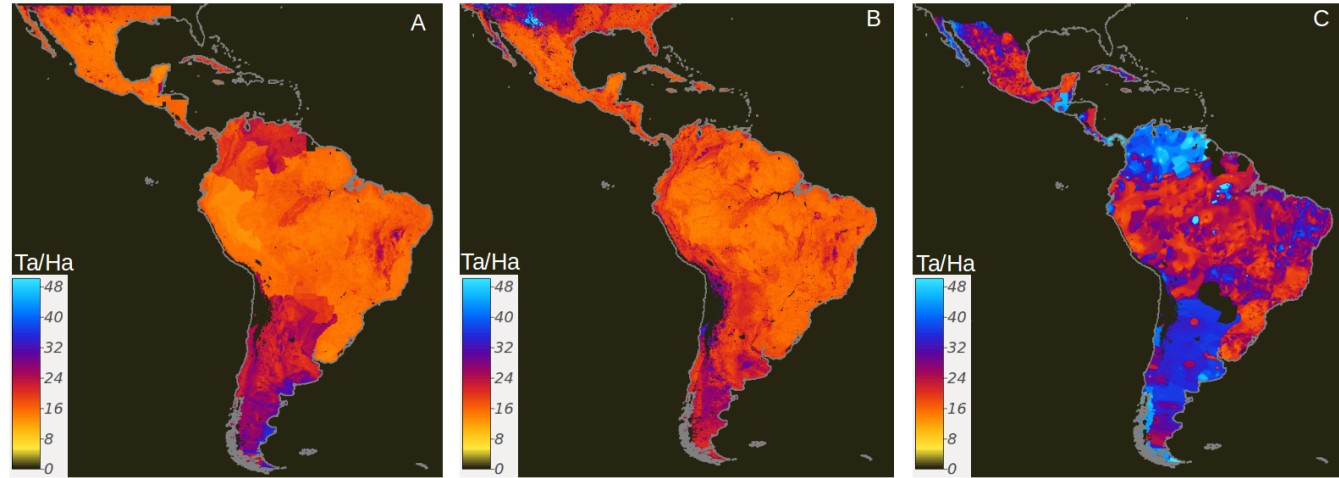

**Figure 6.** In A The full conditional response of residuals to the prediction factors in a country-specific basis. In B the full conditional response of residuals to the SOC prediction factors in the global (Latin America) model. In C the trend of the approximated error of SOC estimates derived from buffer distances and the random forest spatial framework.

predict higher SOC values are larger than otherwise (Fig. 6B). However, we report a similar SOC stock from both modeling approaches.

## 3.7 SOC stocks and model uncertainties

For our model, the uncertainty of the maximum prediction limit was estimated to be $\pm 10$ Pg, which was the variance of the SOC stock by increasing the prediction limit from 1 to 700 Ta/Ha. This relationship showed a stable (close to 0) trend after 200 Ta/Ha. The larger density of extreme values was found with the regional model, and we calculated a maximum possible SOC stock 83.62 Pg from this model.

Despite the spatial differences reported for the country-specific and regional ensembles, we report a similar stocks between both approaches (77.8 $\pm 42.2$ and 76.8 $\pm 45.1$ Pg respectively). We found the global ensemble yielding a slightly higher uncertainty. Our country-specific ensembles are suggesting that countries with highest SOC stocks were Brazil, Argentina, Colombia, Mexico, Peru and Venezuela (Table 3).

Consistently, all models show that Tropical broadleaf evergreen forest, Croplands and Temperate shrublands are the land cover classes that have higher SOC across all SOC available estimates (Table 4). However, using only the dataset contained in the WoSIS system, we predict nearly the half of SOC compared with global models.

The model variance of predicted SOC reached values over 300% for countries such as Mexico and Bolivia. In contrast, countries with higher SOC per unit area and a relatively low prediction variances were Panama, Guatemala, Costa Rica, Nicaragua and Belize. Overall, we found a median model prediction variance of 53% across countries in Latin America. Areas with high uncertainty and model variance were across northern Mexico, Central America, limits between Colombia and Brazil, and the border between Chile and Argentina.

## 4 Discussion

We developed a reproducible DSM framework to characterize the spatial variability of SOC across Latin America. Our results suggest that a multi-model approach is suitable to better understand modeling bias and uncertainty of SOC maps. We argue that uncertainty on SOC mapping can be associated with a) the property of interest (i.e., SOC), b) the environmental complexity and area/country of interest, and c) the characteristics of available data (e.g., spatial distribution and representativeness) to meet model-specific assumptions. Thus, when legacy soil profile collections that were collected for different purposes along large periods of time (i.e., decades) a multi-model approach (i.e., ensemble) would be convenient to maximize, as much as possible, the predictive capacity considering the available information.

To maximize accuracy of our models, we use a simple linear blend of single predictions that at the continental scale was able to explain 39% of SOC variance using only the information available in the WoSIS system. This result falls within the range of the prediction capacity of country-specific models . Besides the low density of observation points, the performance could be partially affected by the generalization from the 1:1 scale of a soil profile (or field SOC observation) to a 5x5km grid, representing an additional source of uncertainty. Higher discrepancy between country-specific and global efforts is evident across

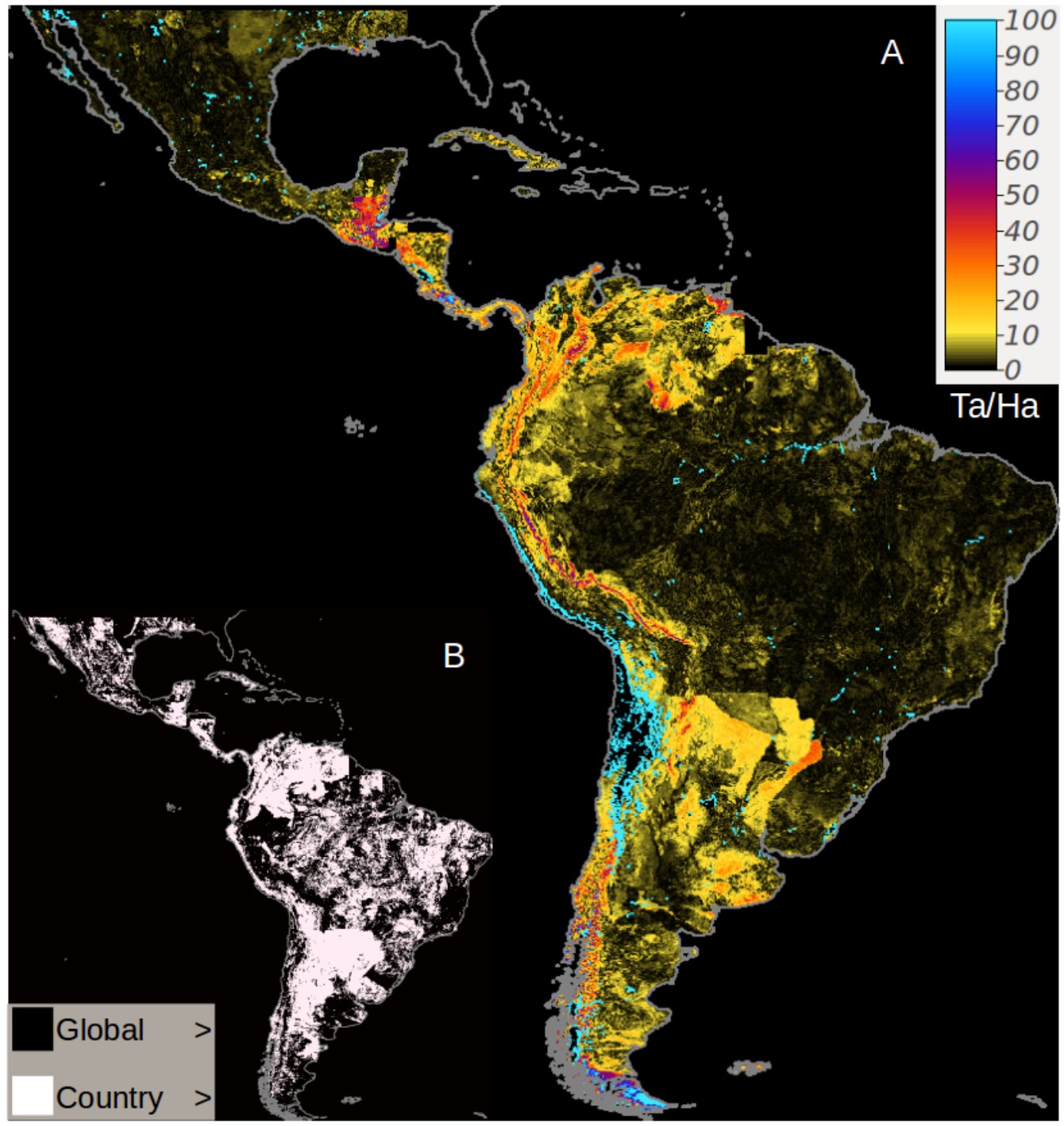

**Figure 7.** In A The absolute distance between the country-specific and the global ensemble. In B the areas in white are areas where the country specific (Country >) is predicting higher carbon than the global ensemble (Global >).

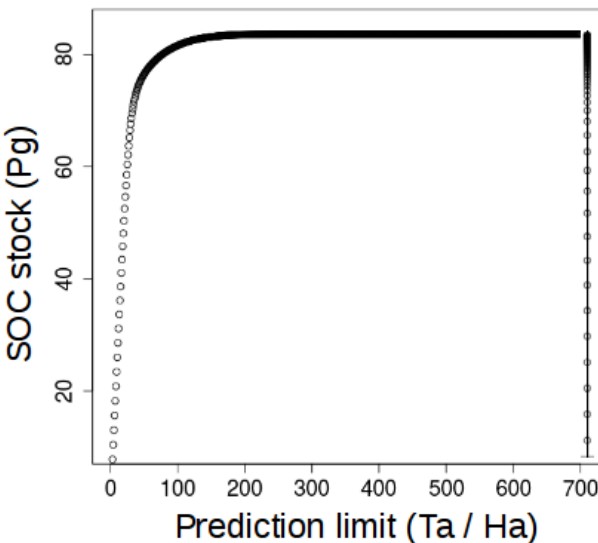

**Figure 8.** The relationship between the SOC stock and the prediction limit. The averaged breakdown points of this relationship are shown in the vertical line at the right of the plot.

Brazil, the largest country, where our models tend to predict nearly the half of SOC than previous efforts. The SoilGrids system tends to predict the highest values while our country-specific ensemble the lowest. The GSOCmap-GSP and our ensembles predicted <100 Pg of SOC across the analyzed countries, while all other products suggest higher stocks 3).

Another source of discrepancy can be associated with the lack of available data to represent the SOC stock at the depth of interest (i.e., -30 cm of mineral soil). The predictive performance of the mass preservative spline to continuously represent the SOC and depth relationships in some cases could be strongly influenced by the lack of observations across highly variable soil profiles, such as those SOC rich agricultural soil profiles constantly transformed for food production purposes, or a volcanic setting. These high levels of missing data lead the trend map of approximated error (Fig. 6), which provides an idea of the uncertainty in the SOC estimates.

The GSOCmap-GSP, for example, was generated in a country-basis, but the amount of SOC observations used for the countries to generate these maps was considerable higher than the available data in the WoSIS system (> 1000000 points). Both of our models predicted more conservative results than the GSOCmap-GSP, which at the same time, the GSOCmap-GSP predicted less SOC than the SoilGrids system and the Harmonized World Soil Database. Respectively, the SoilGrids system relies on a multivariate space suitable to represent the global soil forming environment, however, a model would assume a similar relation of each covariate with the response across all land area in the world. The Harmonized World Soil Database in other hand may be a pedologically sound product, but large areas of Latin America have not been mapped at detailed scales (i.e., larger scales than 1:1 million) and results in a polygon-based approach relying on wide generalizations.

Despite the aforementioned limitations, across Latin America there is an increasing availability of relevant SOC information across site and country-specific regions (e.g., (Reyes-Rojas et al., 2018; Vasques et al., 2016; Angelini et al., 2017; Samuel-Rosa et al., 2015; Angelini et al., 2016; Padarian et al., 2017)) which could serve for validating and calibrating global SOC estimates. Thus, regional approaches considering multiple Latin American countries and SOC models could be a valuable resource to explain discrepancies between site or country-specific and global SOC models.

Our results incorporate a multi-model perspective for quantifying/evaluating the spatial variability of SOC. The model with higher predictive capacity in terms of cross validated $r^2$ was RF, an ensemble of regression trees based on bagging, however this method yields high $EC_r$ and therefore it tends to capture the trend but with high bias. SVM and RK were methods with higher agreement between RMSE and $corr$ and therefore, lower $EC_r$. Large values of $EC_r$ represent an accuracy limitation that was evident for RF, PL, and KK. To overcome these type of modeling biases, previous studies have suggested that the theory of ensemble learning applied to soil datasets could increase the accuracy of results (Finke, 2012; Nussbaum et al., 2018). Furthermore, recent studies highlight the applicability of selective ensembles across a large diversity of model algorithms useful for digital soil mapping purposes (Møller et al., 2018) Thus, our modeling approach includes the combination of multiple predictions by using a linear stack of models as implemented in the caretEnsemble package of R (Deane-Mayer and Knowles, 2016), with the ultimate goal of reducing the uncertainty on SOC mapping efforts.

This study is expected to increase the capacity of Latin American institutions to provide accurate baseline estimates of SOC with a country-specific perspective following recommendations of GSOCmap-GSP. Ultimately, these efforts will enhance the development of new guidelines for measuring, mapping, reporting, verification and monitoring SOC stocks at national level (Vargas et al., 2013). Accurate country-specific DSM frameworks for SOC are required to facilitate interoperability and inform environmental policy across developing countries (Vargas et al., 2017). Our results highlight that attention is needed to better understand the influence of model prediction limits (e.g., the full conditional distribution) for the predicted SOC stocks. Setting an unreliable (excessive or low) prediction limit can have important effects (under or overestimating) on the overall estimated stocks (Table. 3). Therefore, we argue that data science systems for DSM carbon assessments should be fundamentally based on SOC expert knowledge and informed by expert-based soil mapping systems.

Across Latin America we did not find a common predictive algorithm for SOC. These results suggest that country-specific environmental predictors and available data influence the applicability of different approaches. This assessment is needed to address the requirements from the GSOCmap-GSP with the official mandate to generate and update country-specific soil information by the means of DSM. Thus, we argue that the DSM form of each country should assess and incorporate country-specific available data and environmental predictors to select the best prediction algorithm. The FAO SOC mapping cookbook explores possibilities to derive country-specific SOC maps from a variety of prediction algorithms (Yigini et al., 2018), and multiple resources have described the state of the art of modeling methods focused on DSM of soil carbon (Minasny et al., 2013; Malone et al., 2017) including geostatistics (Hengl, 2009, 2017). Thus, data characteristics (e.g., spatial structure, representativeness) are specifically important for developing a DSM framework as legacy soil profile collections, generated with long-term soil inventory purposes, will determine data availability and spatial distribution within a country.

This country-specific approach to map regional SOC results in artifacts across geo-political borders. Therefore, data sharing, model validation and calibration experiments across borders (i.e., between countries) are required to better capture the spatial variability of SOC. The use of a natural-defined prediction domain (e.g., ecoregional or physiographic map) could reduce the border effects. However, we understand that geo-political limits are required for public policy decisions around country-specific needs. We highlight that there is a lack of publicly available country-specific data that ultimately influence the performance of both country-specific and global SOC estimates .

To achieve the highest possible accuracy of country-specific SOC estimates, the availability of point data sources for SOC modeling and mapping is an important consideration selecting an efficient modeling strategy, specially dealing with legacy SOC datasets. Our results highlight important uncertainty levels ( >100%) across large areas of Latin America 6. The data contained in WoSIS has a low-density distribution given the large area and environmental complexity of several analyzed countries. Thus, larger uncertainty dominates countries with larger carbon pools probably because available data does not capture the large spatial heterogeneity of SOC stocks. We highlight that the WoSIS dataset is a unique and invaluable effort that has proven to generate global SOC predictions (Hengl et al., 2017; Sanderman et al., 2017), but there is a global need to increase information and networking capabilities for SOC (Harden et al., 2017).

This study generated predictions of SOC across Latin America, but also provided information about the main relationships driving the spatial distribution of SOC. Machine learning (i.e., data driven) models have proven to be more efficient to model non-linear relationships of SOC (Hengl et al., 2015), but our results suggest that linear-based models (e.g., RK) could outperform machine learning methods under well distributed and representative SOC data scenarios. Similar results were found across productive landscapes of Brazil (Bonfatti et al., 2016). We argue that our capacity to meet modeling assumptions will determine the most suitable prediction algorithm or ensemble methods (i.e., stack, blend, bucket of models). Machine learning models are usually conceived as black boxes and the influence of non-informative SOC prediction factors on machine learning-based SOC models has not been evaluated in detail. Therefore, we propose that the use of simple linear methods (i.e., correlation of available data and its predictors) can be a useful and parsimonious first step to inform data driven approaches and enhance the interpretability of machine learning models to predict SOC. Although ideally, the simple selection of prediction factors based on simple correlation analysis does not prevent multi-collinearity, in which hypothesis driven methods (e.g., RK) may be in risk to fail, other approaches may be able to avoid the bias associated with the statistical redundancy (e.g., PL) or machine learning method where no assumption about multicollinearity exist (e.g., RF). Furthermore, our data suggests that country-specific predictor factors are needed to better parameterize models but also could be useful for country-specific model interpretation. These results have important implications because it has been proposed that an extensive set of prediction factors are required to capture the large variance of the global SOC pool (Hengl et al., 2017). Thus, we propose that a limited but informative country-specific prediction factors should be explored to describe the local biophysical characteristics controlling SOC variability.

## 5 Conclusions

We provide a multi-model comparison approach to map SOC stocks across Latin America and found that there is not a dominant best prediction algorithm given available data. The relatively performance of the different methods vary from one place to another as well as the relatively correlation of SOC with the prediction factors given available data. We compared and combine

hypothesis driven approaches (e.g. linear Geo-statistics) and data driven algorithms (e.g. machine learning) which are used, respectively, to generate interpretable and predictable models of soil variability. We argue that models should not be conceived as competitors, because they have different assumptions (about the data itself, or about the empirical relationship between the response variable and its predictors). Therefore, different models will capture different portions of soil variability. There are no silver bullets on digital soil mapping across the 19 analyzed countries given available data in the WOSIS system. We highlight

potential levels of uncertainty in SOC stocks associated with the maximum allowed prediction limit. Public data may not be representative across large areas and we call for the countries to strength digital soil mapping capacity building initiatives, reproducible research and data sharing. The use country-specific information and the use of different modeling approaches will enhance regional soil carbon mapping efforts, so we can easily identify where and the reasons why different modeling approaches generate different results.

*Code availability.*  The code used for this work will be available under the AGPL 3.0 license at https://github.com/DSM-LAC/NoSilverBulletsForDSM (Guevara et al., 2018)

*Data availability.*  The soil dataset used is this paper is kindly provided by ISRIC-World-Soil-Information. It can be downloaded from WOSIS http://www.isric.org/explore/wosis and correponds to the July 2016 snapshot (Batjes et al., 2017). Soil covariates are available thanks to the ISRIC initiative worldgrids.org

## Appendix A

### A1  Brief description of implemented methods

RK is a hybrid model with both, a deterministic and a stochastic component (Hengl et al., 2004). The regression part took form of a step-wise (back and forward) multiple linear regression to avoid statistical redundancy among the best prediction factors. The residual kriging was ordinary. The variogram parameters supporting the spatial interpolation were automatically

fitted using the framework proposed by Hiemstra et al. (2008). RK was applied only to countries with 10 or more available observations.

PLS is a common method to deal with the presence of highly correlated predictors. The PLS algorithm integrates the compression and regression steps and it selects successive orthogonal factors that maximize the covariance between predictor

and response variables (Wold, 1983; Viscarra Rossel et al., 2014). Most of its development and application is in the fields of chemometrics, but is used in several research areas to effectively solve regression and classification problems.

SVM apply a simple linear method to the data but in a high-dimensional feature space non-linearly related to the input space (Karatzoglou et al., 2006). It creates a hyperplane through n-dimensional spectral-space. Then, SVM separates numerical data
based on a kernel function and parameters (e.g. gamma and cost) that maximize the margin from the closest point to the hyperplane that divides data with the largest possible margin, being the support vectors the points which fall within (Heumann, 2011). Then, linear models are fitted to the support vectors.

RF is an ensemble of regression trees based on bagging (Breiman, 1996). This machine learning algorithm uses a different combination of prediction factors to train multiple regression trees. Each tree is generated using a different subsets of available
data (Breiman, 2001). The number of prediction factors to use on each tree is known as the mtry parameter. The final prediction is the weighted average of all individual trees.

KK is a pattern recognition technique which is based on the distances to training examples in the feature space (Silverman and Jones, 1989). The observations within the learning set, which are particularly close to the new observation (y, x), should get a higher weight in the decision than such neighbors that are far away from (y, x) (Hechenbichler and Schliep, 2004).
The parameter k determines the number of neighbors from which information will be considered for prediction and a kernel function (eg. triangular, Gaussian among others) converts distances into weights which will be used for regression problems.

*Competing interests.* The authors declare that they have no conflict of interest.

*Acknowledgements.* This work was supported by the Global Soil Partnership, the Central America, Caribbean and Mexico Soil Partnership and the South America Soil Partnership in collaboration with the Department of Plant and Soil Sciences at the University of Delaware.
MG acknowledges support from a Conacyt fellowship. GFO is supported by the Argentinian government through the project INTA PN-SUELO1134032. RV acknowledges support from NASA (80NSSC18K0173) and USDA (2014-67003-22070).

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

**Table 2.** Best correlated predictors and its frequency across the analyzed data country-scenarios, given available data in the WOSIS system. See the predictor codes in http://worldgrids.org/doku.php/wiki:layers. ARG=Argentina, BLZ=Belize, BOL=Bolivia, BRA=Brazil, CHL=Chile, COL=Colombia, CRI=Costa Rica, CUB=Cuba, DOM=Dominican Republic, ECU=Ecuador, ESP=Espana, GTM=Guatemala, HND=Honduras, JAM=Jamaica, MEX=México, NIC=Nicaragua, PAN=Panama, PER=Peru, SUR=Suriname, SLV=El Salvador, URY=Uruguay, VEN=Venezuela

| Var | factor | subfactor | freq | Country |
|---|---|---|---|---|
| gachws3a | Soil | Soil type | 2 | CUB, SUR |
| garhws3a | Soil | Soil type | 2 | PER, URY |
| ghshws3a | Soil | Soil type | 2 | BLZ, URY |
| gphhws3a | Soil | Soil type | 2 | CUB, JAM |
| gplhws3a | Soil | Soil type | 2 | BLZ, BOL |
| gvrhws3a | Soil | Soil type | 2 | JAM, URY |
| tdmmod3a | Climate | Temperature | 11 | ARG, BOL, BRA, CHL, COL, CRI, CUB, ECU, MEX, PER, VEN |
| tx1mod3a | Climate | Temperature | 10 | ARG, BOL, BRA, COL, CUB, ECU, JAM, NIC, PER, URY |
| tx4mod3a | Climate | Temperature | 10 | BRA, CHL, CRI, CUB, ECU, GTM, JAM, MEX, PER, VEN |
| tx5mod3a | Climate | Temperature | 9 | BOL, BRA, CHL, CUB, ECU, JAM, MEX, PER, VEN |
| tx6mod3a | Climate | Temperature | 9 | ARG, BOL, BRA, CHL, COL, CRI, ECU, MEX, VEN |
| tnhmod3a | Climate | Temperature | 8 | BLZ, COL, CRI, GTM, HND, JAM, PAN, VEN |
| tnmmod3a | Climate | Temperature | 8 | BLZ, COL, CRI, GTM, HND, PAN, URY, VEN |
| tx3mod3a | Climate | Temperature | 7 | BRA, CHL, CUB, ECU, PAN, PER, VEN |
| tdhmod3a | Climate | Temperature | 6 | ARG, CUB, ECU, JAM, MEX, URY |
| tdlmod3a | Climate | Temperature | 6 | BRA, CHL, COL, ECU, GTM, JAM |
| tnsmod3a | Climate | Temperature | 5 | ARG, MEX, NIC, PAN, SUR |
| tx2mod3a | Climate | Temperature | 4 | ARG, ECU, PER, URY |
| tdsmod3a | Climate | Temperature | 3 | MEX, PAN, SUR |
| tnlmod3a | Climate | Temperature | 3 | BLZ, COL, GTM |
| px2wcl3a | Climate | Precipitation | 2 | BOL, PAN |
| px3wcl3a | Climate | Precipitation | 2 | CHL, MEX |
| px4wcl3a | Climate | Precipitation | 2 | BRA, CHL |
| etmnts3a | Climate | ET | 2 | ARG, MEX |
| evmmod3a | Organism | Vegetation | 5 | ARG, ECU, HND, MEX, VEN |
| l07igb3a | Organism | Vegetation | 2 | ARG, CHL |
| DEMSRE3a | Topography | | 5 | COL, CRI, GTM, HND, SUR |
| twisre3a | Topography | | 5 | BRA, JAM, NIC, PAN, SUR |
| ChannNetworkBLevel | Topography | | 4 | COL, HND, PAN, SUR |
| l3pobi3b | Topography | | 4 | COL, CRI, PAN, VEN |
| inssre3a | Topography | | 3 | BLZ, HND, SUR |
| opisre3a | Topography | | 3 | CRI, NIC, SUR |
| SLPSRT3a | Topography | | 3 | CRI, NIC, SUR |
| AnalyticalHillshading | Topography | | 2 | BLZ, CUB |
| Aspect | Topography | | 2 | BLZ, BOL |
| CovergenceIndex | Topography | | 2 | BOL, HND |
| inmsre3a | Topography | | 2 | CRI, GTM |
| ValleyDepth | Topography | | 2 | BLZ, JAM |
| geaisg3a | Age | | 3 | CHL, NIC, SUR |

**Table 3.** SOC stocks (Pg) at the contextual resolution of 5x5km grids. ens = country-specific, global=Latin America ensemble, sg= SoilGrids system, GSOCmap-GSP= country-specific 1km, hw=Harmonized Word Soil Data Base.

| | country | ens | global | sg | GSOCmap-GSP | hw |
|---|---|---|---|---|---|---|
| 1 | ARGENTINA | 13.19 | 12.77 | 24.45 | 18.00 | 18.13 |
| 2 | BELIZE | 0.24 | 0.12 | 0.28 | 0.28 | 0.19 |
| 3 | BOLIVIA | 3.29 | 3.39 | 8.39 | 6.99 | 5.96 |
| 4 | BRAZIL | 26.82 | 27.16 | 68.45 | 42.79 | 47.20 |
| 5 | CHILE | 6.31 | 7.20 | 15.15 | 1.93 | 8.28 |
| 6 | COLOMBIA | 7.01 | 5.96 | 15.50 | 5.12 | 14.99 |
| 7 | COSTA RICA | 0.56 | 0.34 | 0.83 | 0.83 | 0.71 |
| 8 | CUBA | 0.52 | 0.51 | 1.48 | 0.82 | 0.64 |
| 9 | ECUADOR | 1.31 | 1.36 | 4.04 | 1.57 | 2.63 |
| 10 | GUATEMALA | 1.02 | 0.57 | 1.27 | 1.27 | 0.99 |
| 11 | JAMAICA | 0.05 | 0.05 | 0.14 | 0.07 | 0.07 |
| 12 | MEXICO | 5.98 | 6.12 | 14.43 | 9.04 | 17.59 |
| 13 | NICARAGUA | 0.74 | 0.62 | 1.42 | 0.71 | 0.92 |
| 14 | PANAMA | 0.56 | 0.43 | 1.10 | 0.33 | 0.69 |
| 15 | PERU | 4.38 | 5.13 | 17.08 | 3.14 | 10.51 |
| 16 | SURINAME | 0.56 | 0.51 | 1.20 | 0.45 | 1.33 |
| 17 | URUGUAY | 0.92 | 0.88 | 1.99 | 0.84 | 2.27 |
| 18 | VENEZUELA | 4.71 | 3.77 | 9.39 | 5.28 | 5.64 |

**Table 4.** SOC stocks at the contextual resolution of 5x5km across Land cover classes of Latin America for the 18 analyzed countries. ens = country-specific, global=Latin America ensemble, sg= SoilGrids system, GSOCmap-GSP= country-specific 1km, hw=Harmonized Word Soil Data Base.

|  | land cover | ens | GSOCmap-GSP | hw | sg | global |
|---|---|---|---|---|---|---|
| 1 | Tropical broadleaf evergreen forest | 30.39 | 40.30 | 59.15 | 80.44 | 29.73 |
| 2 | Tropical broadleaf deciduous forest | 0.43 | 0.65 | 1.00 | 1.09 | 0.42 |
| 3 | Sub-tropical broadleaf evergreen forest | 2.38 | 3.91 | 4.51 | 6.57 | 2.25 |
| 4 | Sub-tropical broadleaf deciduous forest | 1.42 | 2.04 | 1.87 | 2.55 | 1.07 |
| 5 | Temperate broadleaf evergreen forest | 3.32 | 1.26 | 4.97 | 6.91 | 3.56 |
| 6 | Temperate broadleaf deciduous forest | 0.48 | 0.52 | 1.02 | 1.21 | 0.63 |
| 7 | Sub-tropical needleleaf forest | 0.00 | 0.01 | 0.00 | 0.01 | 0.00 |
| 8 | Temperate needleleaf forest | 0.23 | 0.36 | 0.45 | 0.54 | 0.24 |
| 9 | Mixed forest | 0.67 | 1.08 | 1.34 | 1.66 | 0.66 |
| 10 | Tropical shrubland | 4.25 | 6.58 | 6.98 | 10.30 | 4.18 |
| 11 | Sub-tropical shrubland | 3.17 | 4.18 | 6.62 | 6.33 | 2.90 |
| 12 | Temperate shrubland | 4.56 | 5.08 | 7.33 | 9.97 | 5.32 |
| 13 | Tropical grassland | 3.01 | 2.48 | 3.56 | 5.46 | 2.45 |
| 14 | Sub-tropical grassland | 1.15 | 1.35 | 2.28 | 2.58 | 1.12 |
| 15 | Temperate grassland | 2.75 | 3.31 | 4.86 | 5.92 | 3.04 |
| 16 | Water | 1.21 | 1.37 | 2.07 | 3.45 | 1.21 |
| 17 | Urban area | 0.24 | 0.31 | 0.45 | 0.55 | 0.22 |
| 18 | Permanent ice and snow | 0.14 | 0.08 | 0.14 | 0.38 | 0.17 |
| 19 | Barren land | 1.74 | 2.38 | 2.43 | 2.95 | 1.70 |
| 20 | Cropland | 12.95 | 19.33 | 21.89 | 27.94 | 12.42 |
| 21 | Wetland | 0.37 | 0.56 | 0.66 | 1.24 | 0.35 |
| 22 | Salt flat | 0.13 | 0.17 | 0.16 | 0.18 | 0.10 |
| 23 | Sea water | 1.59 | 1.39 | 2.23 | 4.31 | 1.78 |