# Peer review of "No Silver Bullet for Digital Soil Mapping: Country-specific Soil Organic Carbon Estimates across Latin America"

_SOIL, 2017_

## Referee Comment (RC1) · T. Hengl (Referee) · 7 Mar 2018

The paper compares five statistical/machine learning techniques for the purpose of mapping organic carbon stock 0-30 cm for 19 countries in Latin and Central America: SVM, RF, PL, KK and RK. The authors conclude, based on their results, that (P14L13) "there are no silver bullets on digital soil mapping" meaning that no single technique outperforms other in sense of accuracy. Also, authors further conclude that countries need to work on improving the quality and quantity of ground data on soil carbon. Methodology is relatively well explained and I especially appreciate that most of the code is also available for review (https://github.com/DSM-

LAC/NoSilverBulletsForDSM), which is really the best way to publish scientific work. Although a valuable piece of work that brings >20 soil data producers from Latin America, I have some questions concerning the methodology and results (some sections lack clarity and especially methods section needs to be extended), and also some reservations considering the main messages of the article. I hope that the authors (and the readers) will recognize some of these points and that these will help them improve their paper:

1. Weighted overall measure of mapping accuracy maybe more informative than stats per country? The main result of the paper is that neither of the five methods considered (SVM, RF, PL, KK and RK) results in significantly more accurate results. I have however two concerns about the P10L18-20 and Figure 2: (a) since two countries (BRA and MEX; Table 1) have over ca 5 times more points than all other countries together, I think it would be more fair to compare overall accuracy of methods using a weighted overall measure of mapping accuracy (where the weight could be either size of country, or better number of points per country), (b) another weighting factor that could be used is the accuracy of estimated soil carbon stocks (t/ha) since different points come with different accuracy (success of fitting splines to irregular soil horizon data can be estimated so that this information can be provided per point). I believe that providing a weighted overall measure of mapping accuracy for the five methods would give a more objective view of which method is more accurate. At the moment I see that (P10L18) for BRA the best method is RF and for MEX RK, hence these would be clear winners considering that these are based on ca 10,000 points (>85% of all points).

2. Building models with <30 training points can lead to artifacts Table 1 indicates that, from 19 countries, only 7 countries have >100 points available for modeling. Modeling and comparing models for countries with <30 points I would not even recommend as fitting of variograms for data sets with <50 is rather tricky and can lead to artifacts. Oliver and Webster (2014; https://doi.org/10.1016/j.catena.2013.09.006) suggest that one should collect at least 100-150 samples to get a reliable estimates of variogram

parameters. I would not be that strict but at least, any results you get for countries with <30 points should probably be critically evaluated.

4. Is local better than global? Is it justified to stitch maps produced by countries vs using global models? Authors provide comparison of state-of-the-art methods for generating spatial predictions using soil carbon stocks and large stack of environmental covariates. However, I firmly believe that most of the readers would in fact be more interested in finding out whether (a) building N local models per country and then making predictions, or (b) fitting a single global model using all data, is more accurate? So adding a section 3.4b "SOC predictions based on global models" and predicted values to the plot Figure 3, would probably significantly increase value of this paper. Also, it would prove that the FAO GSP's choice to let countries map properties and then stitch maps together, is a better option than to merge all points together and then fit single global models. Read more about this discussion in http://www.pedometrics.org/Pedometron/Pedometron38.pdf "On usability of soil maps (and on global soil data models vs stitching together of individual disparate soil maps)"

5. Evaluation of the accuracy of predictions should ideally be based on e.g. k-fold CV with re-fitting It is not entirely clear from the manuscript how was the model evaluation implemented (section 2.3). I would expect a 5-fold CV with model refitting i.e. "repeatedcv" (https://topepo.github.io/caret/model-training-and-tuning.html#basic-parameter-tuning) - is this the one you used? Note that "repeatedcv" ensures that (a) models are repeatedly fitted and (b) there is enough repetition to get stable results. Please provide half page explaining how exactly is the accuracy / RMSE derived.

6. Github repository missing training data Github repository does not contain all points and grids you have downloaded from ISRIC WoSIS and worldgrids.org. I would at least appreciate if you could put the main regression matrix containing values of the target variable and all covariates. This way we would be able to reproduce your results, as in Nussbaum et al. (2018; https://doi.org/10.5194/soil-4-1-2018).

[Figure]

On the end, I should also mention that I am big supporter on connecting countries and especially researchers and applied specialists in countries around the world to share data on soil (and this paper clearly contributes to this initiative). I am not as big supporter of making issues such as climate change, deforestation, soil erosion, soil carbon and similar, become questions of national sovereignty and/or political debate. Or to quote Neil deGrasse Tyson: "Objective truths are established by evidence. Personal truths by faith. Political truths by incessant repetition." See also: https://www.facebook.com/neildegrassetyson/videos/10155195888806613/ and that is why I have especially high reservations towards letting countries freely choose the "most accurate" method to determine soil carbon stocks.

Other detailed comments in the manuscript and questions are available in the appendix (PDF).

Yours,

T. Hengl https://orcid.org/0000-0002-9921-5129

Please also note the supplement to this comment:
https://www.soil-discuss.net/soil-2017-40/soil-2017-40-RC1-supplement.pdf
* * *
[Figure]

**Supplement:**

[revised manuscript text omitted]

1. **support vector machines, random...**
use consistently abbreviations for each method throughout the paper e.g. support vector machines (SVM), regression-kriging (RK) etc; *[tom.hengl]*

2. **ISRIC-World-Soil-Information-System.**
ISRIC - World Soil Information institute *[tom.hengl]*

3. **(5x5km pixel resolution)**
We could provide you also with 250 m resolution data - for sure working with very coarse resolution results in higher uncertainty of predictions. The FAO GSOC map is provided at 1 km, so somewhat pity to see that you use only very coarse resolution. *[tom.hengl]*

4. **data scenarios**
what is a "data scenario"? Please clarify. *[tom.hengl]*

5. **had lower SOC stocks per unit area**
hm, why would be the mean stocks be lower in larger countries? *[tom.hengl]*

6. **We highlight that setting unreliable...**
I am still not sure about what do mean here - what are model prediction limits? You predict and then you filter based on some physical limits to avoid overshooting? *[tom.hengl]*

7. **The overarching goal is that a...**
Is this decision based on thorough comparison of methods and approaches? What is global models result in higher accuracy than a compilation of local models? *[tom.hengl]*

[revised manuscript text omitted]

1. **Arguably, there are two cultures...**
All well known. Inserting, however, here differences between doing global vs local models would be more interesting. *[tom.hengl]*

2. **We argue that a systematic analysis...**
Fully agree. *[tom.hengl]*

[Figure]

[Figure]

carbon models (Martin et al., 2011; Hashimoto et al., 2017; Hengl et al., 2017) including applications for SOC mapping (Grimm et al., 2008; Sreenivas et al., 2016; Yang et al., 2016; Hengl et al., 2017; Delgado-Baquerizo et al., 2017; Ließ et al., 2016; Viscarra Rossel et al., 2014).Machine learning methods do not necessarily allow to extract information about the main effects of prediction factors in the response variable (e.g., SOC); consequently, a selection strategy is always useful to increase

5    the interpretability of machine learning algorithms. With this diversity of approaches one constant question is if there is a method that systematically improve the prediction capacity of the others aiming to predict SOC across large geographic areas (e.g., Latin America). We postulate that probably there is no universal method (i.e., silver bullet) for DSM, and country-specific efforts are needed to test a variety of predictive algorithms to maximize explained variance while minimizing prediction bias.

   The overarching goal of this study is to compare different predictive algorithms across 19 data/country scenarios with

10   publicly available information to support the development of country-specific SOC maps to be included in the GSOCmap-GSP. Currently, SOC information across Latin America is derived from global models such as the SoilGrids system, or the Harmonized World Soil Database (Hengl et al., 2017; Köchy et al., 2015), which lack quantification of uncertainty and large areas are parameterized with limited country-specific information. This challenge is not unique for Latin America as many regions around the world (e.g., Africa, Siberia) have limited SOC information to parameterize models to predict SOC. To

15   inform future SOC mapping efforts, this study addresses two specific questions: a) Which environmental variables (derived from publicly available information) have the highest correlations with country-specific SOC information?; and b) Which is the best method (i.e., predictive algorithm) to represent SOC across Latin America and within each country? The ultimate aim of this study is to contribute with the discussion about the importance of integrating country-specific information for representing and predicting soil-related variables (e..g., SOC) to improve regional-to-global predictions.

20   ## 2   Methods

**2.1   SOC observations**

Soil organic carbon information was extracted from the WoSIS soil profile database . This dataset includes local-to-national soil profile collections with a sampling strategy generally based on morphological soil attributes (Batjes et al., 2017). The goal of the GSOCmap-GSP is to produce global information for the first 30 cm; thus, we generated synthetic horizons for this depth

25   using a mass preserving spline approach (Bishop et al., 1999). We applied a pedotransfer function if the bulk density (BLD) information was missing (Yigini et al., 2017), and assumed a value of 0% of coarse fragments when information on coarse fragments (CRFVOL) was missing. The organic carbon stock for 0 to 30 cm was estimated using Global Soil Information Facilities R, GSIF following a standardized SOC calculation method (D.W. and L.E., 1982):

$$SOC_{stock} = \frac{ORC}{1000} \times \frac{H}{100} \times BLD \times \frac{(100 - CRFVOL)}{100} \qquad (2)$$

30   where $ORC$ is SOC density ($g \cdot kg^{-1}$) and $H$ is soil depth (30 cm). Finally, each country-specific dataset was transformed to its natural logarithm to reduce the right-skewed distribution of SOC values and because exploratory analysis showed that this
* * *
**Margin notes:**

1. **We postulate that probably there is...**
Mapping SOC is not = DSM. SOC is just one of the properties. Organic carbon stock is just on of the SOC variables. *[tom.hengl]*

2. **The ultimate aim of this study...**
Again, what would be very interesting in this context that you provide evidence that for example many local models will in average be more accurate than one global (whole Latin America) model. *[tom.hengl]*

3. **database .**
database. *[tom.hengl]*

4. **was extracted from the WoSIS soil...**
I am amazed that these countries did not contribute their own data. For sure countries such as Argentina Chile etc have much much more points than those that are available via ISRIC WISE etc. *[tom.hengl]*

5. **assumed a value of 0% of coarse...**
Both using PTF and fitting splines introduces uncertainty. Please provide more info in the results and discussion. *[tom.hengl]*

6. **SOC**
The key is to make distinction between the four SOC variables here (content, density, stock and total stock). See http://gsif.isric.org/doku.php/wiki:soil_organic_carbon *[tom.hengl]*

transformation can improve the prediction capacity of further modeling methods. To analyze the statistical distribution of SOC values, a probability distribution function was plotted and a Shapiro-Wilk test of normality was conducted on each dataset.

**2.2 Soils prediction factors**

We used environmental information from WorldGrids (worldgrids.com), which is an initiative of ISRIC-World Soil Informa-
5  tion. We downloaded and masked 118 environmental layers (i.e., prediction factors) for each country to quantitatively represent the soil forming environment. The prediction factors were harmonized into a 1x1km global grid by the WorldGrids project from three main information sources: remote sensing, climate surfaces, and digital terrain analysis (http://worldgrids.org/doku.php/wiki:layers). Additional terrain parameters (e.g., terrain slope, aspect, catchment area, channel network base level, terrain curvature, topographic wetness index, length-slope factor) from elevation data were calculated in SAGA GIS for each country following the
10  standard implementation for basic terrain parameters (Conrad et al., 2015). We re-sampled the prediction factors into a 5x5km pixel size grid to reduce the computational demand required to make predictions and facilitate the reproducibility of this DSM framework without the need of High Performance Computing.

**2.3 Prediction of SOC and model evaluation**

First, the relationship between SOC and prediction factors was explored using simple correlation analysis. Second, the 10
15  prediction factors with highest correlations with SOC data were selected for each country and used for further analyses. Third, we implemented Regression-Kriging (based on a multiple linear regression model (RK) and partial least squares regression (PLS)), and three machine learning models: support vector machines (SVM), random forests (RF), and kernel weighted nearest neighbors (KK) to generate SOC maps for each country. A brief explanation for each modeling approach is provided in Appendix A1.

20     We also analyzed the influence of the maximum allowed prediction limits for each prediction algorithm. The units of the SOC estimates are $kg \cdot m^{-2}$. The sensitivity of the total SOC stock related to the model prediction limit was tested by changing the maximum prediction limit from $2.7 \, kg \cdot m^{-1}$ ( 1 in a log scale) to $2980.95 \, kg \cdot m^{-2}$ (8 in a log scale).

    To generate a combined SOC map, we used a weighted average of the country-specific predictions. The weights of this average were defined by the relationship between the errors (measured as the RMSE) and the correlation ($EC_r$). We propose this
25  $EC_r$ as an approach to better understand the agreement between the correlation (calculated by the means of cross validation) and the RMSE (derived from the unbiased residuals of cross validation). Before calculating the RMSE/correlation ratio, the RMSE and the correlation between observed and predicted were standardized (by its maximum and minimum values) to a range between 0 and 1 using:

$$\text{RMSE}_{std} = \frac{\text{RMSE}_i - min(\text{RMSE})}{range(\text{RMSE})} \tag{3}$$

30

$$\text{corr}_{std} = \frac{\text{corr}_i - min(\text{corr})}{range(\text{corr})} \tag{4}$$
* * *
**Margin comments:**

1. **(http://worldgrids.org/doku.php/wiki:layers)....**
we have much more detailed and up-to-date covariates for you (250m). *[tom.hengl]*

2. **model evaluation**
Have you considered doing any 5-fold or similar CV? I think the readers main interest will be in the accuracy based on CV with re-fitting of the models. I see you are using the caret package but I do not see any model refitting. *[tom.hengl]*

3. **implemented Regression-Kriging...**
itemize - nicer to follow *[tom.hengl]*

4. **The sensitivity of the total SOC...**
I actually do not understand what you are doing here. It looks like some of your algorithms are extrapolating and resulting in very high values that are physically not possible. *[tom.hengl]*

5. **SOC estimates are kg · m −2**
for 0-30 cm *[tom.hengl]*

6. **2980.95**
this number is physically impossible! this would require a bulk density of 5000 kg/m3 or something like this. *[tom.hengl]*

7. **−1**
-2 *[tom.hengl]*

8. **we used a weighted average of the...**
Is this a weighted average of 5 methods or? I am confused with this section. How was exactly the map in Figure 3 generated? *[tom.hengl]*

9. **range between 0 and 1 using:**
Does this comes from some described method in the literature or is this your original invention? I think R-square does exactly what you do here. *[tom.hengl]*

[Figure]

$$EC_r = \frac{\text{RMSE}_{std}}{\text{corr}_{std}} \qquad (5)$$

Where $EC_r$ is the proposed ratio between errors and correlation between observed and predicted (derived by cross-validation); $\text{RMSE}_i$ is the observed RMSE for the $i$th model; $min(\text{RMSE})$ is the minimum observed value of RMSE, and $range(\text{RMSE})$ is the difference between the maximum and minimum observed values of RMSE; $\text{corr}_i$ is the observed correlation for the $i$th model; $min(\text{corr})$ is the minimum observed value of correlation, and $range(\text{corr})$ is the difference between the maximum and minimum observed values of correlation

If the value of the $EC_r$ was close to 0, then there is a stronger agreement between high RMSE and low correlation, or low RMSE and high correlation. If this value deviated from 0 (up to 1 or more), then the RMSE would tend to be high while the correlation was also high, suggesting that the method represents the variability of SOC but with high bias. Finally, the uncertainty (represented by the variance of the different prediction approaches) was divided by the mean and multiplied by 100 to provide an interpretable standardized visualization of uncertainty (i.e., in percent). Country-level SOC stocks are reported as the sum of all 5x5km pixels of all SOC predicted values (i.e., weighted average of SOC) within each country. All analyzes were performed using the R software. (R Core Team, 2017).

**3 Results**

**3.1 Descriptive statistics**

SOC across the different countries showed a wide diversity of data-scenarios (Table 1). Costa Rica (with a mean of 11.05 $\text{g} \cdot \text{kg}^{-1}$), Chile (with a mean of 9.88 $\text{g} \cdot \text{kg}^{-1}$) and Colombia (with a mean of 8.15 $\text{g} \cdot \text{kg}^{-1}$) are the countries with the highest SOC values. Brazil (n=5616) and Mexico (n=4321) were the countries with highest data availability. In contrast, Honduras (n=11), Guatemala (n=20) and Belize (n=21) were the countries with less density of of SOC estimated values (Table 1). With the original (untransformed) dataset, the only countries that showed a normal distribution after the Shapiro- Wilk test of normality with an alpha of 0.05 were Belize, Guatemala, Honduras and Suriname.

**3.2 Correlation of SOC and its predictors**

Best correlated predictors were not the same across countries. We found higher correlations with the original data sets transformed to its natural logarithm, as data had a right-skewed distribution and did not follow a normal distribution (i.e., log-normal). Highest correlations of available SOC data and its environmental predictors were associated with temperature-related-variables across Honduras, Costa Rica, Peru, Chile, Guatemala and Suriname (the $r^2$ varied from from 0.35 to 0.58). However, there were a low number of available SOC observations across these countries in the WoSIS system (between 11 to 34). Similarly, across countries with high data availability (e.g., Mexico and Brazil) the strongest correlations between SOC and prediction factors were associated with temperature-related variables (Table 2). In all cases, the relationship between SOC

1. **Finally, the 10 uncertainty (represented...** This is just variance of different predictions. This is NOT the actual uncertainty of the map in Figure 3 left. If you would add pure noise to the list of predictions that would increase the variance but the actual uncertainty of predictions could be much lower. *[tom.hengl]*

2. **R software. (R** R software (R *[tom.hengl]*

3. **3.2** Here I think you miss a section: "Preparation of point data SOC stocks" describing success of fitting splines to point data (how many points had enough horizons? what is the accuracy of SOC stocks at points? etc). Showing a map of all points used with values for SOC stocks in kg/m2 would also be a good idea. *[tom.hengl]*

[Figure]

[Figure]

**Table 1.** Descriptive statistics of SOC estimates $kg \cdot m^2$ and total land area for each analyzed country. N is the number of observations. We provide quantiles, median, mean and the standard deviation of SOC data. The columns p and plog represent the probability values derived from the Shapiro-Wilk test of normality before (p) and after (plog) the log transformation of SOC values. When p is larger than plog, the log transformation of the data did not increased the probability of normality in the dataset. ARG=Argentina, BLZ=Belize, BOL=Bolivia, BRA=Brazil, CHL=Chile, COL=Colombia, CRI=Costa Rica, CUB=Cuba, ECU=Ecuador, ESP=Espana, GTM=Guatemala, HND=Honduras, JAM=Jamaica, MEX=México, NIC=Nicaragua, PAN=Panama, PER=Peru, SUR=Suriname, SLV=El Salvador, URY=Uruguay, VEN=Venezuela.

| Country | n | Land Area (km2) | Min | 1st Q | Med | Mean | 3rd Q | Max | SDev | p / plog |
|---------|------|-----------------|------|-------|------|-------|-------|--------|-------|-----------------|
| ARG | 231 | 2736690 | 0.34 | 1.88 | 3.21 | 5.65 | 5.96 | 86.85 | 9.33 | <0.001 / 0.03 |
| BLZ | 21 | 22970 | 1.84 | 4.49 | 6.72 | 7.71 | 9.99 | 19.48 | 4.32 | 0.08 / 0.99 |
| BOL | 76 | 1083301 | 0.64 | 1.83 | 2.56 | 2.64 | 3.20 | 7.65 | 1.21 | <0.001 / 0.08 |
| BRA | 5616 | 8358140 | 0.07 | 1.99 | 2.67 | 3.23 | 3.34 | 573.76 | 9.18 | <0.001 / <0.001 |
| CHL | 44 | 743812 | 0.43 | 3.58 | 5.19 | 9.88 | 16.52 | 31.87 | 8.86 | <0.001 / 0.01 |
| COL | 166 | 1038700 | 0.66 | 3.44 | 5.78 | 8.15 | 9.95 | 52.62 | 7.35 | <0.001 / 0.96 |
| CRI | 43 | 51060 | 2.27 | 4.07 | 7.23 | 11.05 | 10.85 | 82.57 | 14.90 | <0.001 / 0.001 |
| CUB | 48 | 109820 | 0.36 | 2.85 | 3.61 | 4.32 | 5.73 | 10.98 | 2.23 | 0.004 / <0.001 |
| ECU | 77 | 276841 | 0.99 | 2.37 | 3.65 | 5.15 | 4.36 | 24.36 | 5.15 | <0.001 / <0.001 |
| GTM | 20 | 107159 | 2.60 | 5.66 | 8.48 | 7.73 | 9.75 | 12.41 | 3.11 | 0.14 / 0.007 |
| HND | 11 | 111890 | 2.69 | 5.25 | 6.48 | 6.71 | 8.32 | 12.38 | 2.78 | 0.72 / 0.39 |
| JAM | 76 | 10831 | 1.29 | 3.01 | 3.99 | 4.35 | 4.83 | 12.90 | 1.99 | <0.001 / 0.72 |
| MEX | 4321 | 1943945 | 0.00 | 1.73 | 2.49 | 2.56 | 3.25 | 35.55 | 1.49 | <0.001 / <0.001 |
| NIC | 26 | 119990 | 2.93 | 3.94 | 7.31 | 7.50 | 9.04 | 15.91 | 3.78 | 0.05/0.09 |
| PAN | 25 | 74177 | 3.39 | 4.90 | 7.53 | 7.59 | 9.13 | 19.89 | 3.76 | 0.003 / 0.49 |
| PER | 145 | 1279996 | 0.19 | 1.89 | 2.93 | 2.92 | 3.55 | 8.35 | 1.42 | 0.005 / <0.001 |
| SUR | 27 | 156000 | 1.38 | 2.60 | 3.35 | 3.37 | 4.07 | 6.01 | 1.20 | 0.69 / 0.51 |
| URY | 130 | 175015 | 0.82 | 2.70 | 3.38 | 4.34 | 3.90 | 46.54 | 4.67 | <0.001 / <0.001 |
| VEN | 164 | 882050 | 0.31 | 2.58 | 4.14 | 5.92 | 6.57 | 44.35 | 6.37 | <0.001 / 0.11 |

1. **5616**
Can you also please add a column with inspection density? i.e. profiles per 100 km2 *[tom.hengl]*

[revised manuscript text omitted]

1. **rmse for each country:** itemize? *[tom.hengl]*

2. **BRA (RF, RF),** Brazil and Mexico have >80% of points. *[tom.hengl]*

3. **The relationship between SOC predicted...** From plot Figure 4B this negative relationship is not very clear. Otherwise yes I would expect higher average errors in countries where average SOC content is somewhat higher. *[tom.hengl]*

4. **SOC stocks and the area of each country** Is it the total stock or average stock? If it is the total stock than this is more than obvious since Area is in the formula for calculating total stocks. *[tom.hengl]*

[Figure]

[Figure]

[Figure]

1. Very difficult to read. I am
sorry but I just do not see
clearly which methods are
best and which are worst.
Too many colors used also
probably. *[tom.hengl]*

[revised manuscript text omitted]

* * *
1. **There are no silver bullets on...**
Also Nussbaum et al. (2017) conclude that model average or ensemble is probably the best way to go as it indeed still improves slightly accuracy.   *[tom.hengl]*

2. **The use country-specific information**
use of   *[tom.hengl]*

3. **at a regional level.**
also at global level?
*[tom.hengl]*

[Figure]

[Figure]

**Figure 4.** SOC stocks and uncertainties. In A we show the strong linear relationship between SOC stocks and the area of each country.[1]
In B we show the negative relationship between the estimated prediction variance (uncertainty component) and the SOC stocks per unit of
area (density of SOC stock). Note how larger uncertainty and lower SOC stocks are associated with larger countries such as Mexico, Brasil,
Bolivia or Peru.

1. **In A we show the strong
linear...**
I wonder if such plot is
needed at all (especially if
you are showing relationship
between the total stock in tC
vs area). *[tom.hengl]*

[Figure]

[Figure]

**Appendix A: Brief description of implemented methods**

[revised manuscript text omitted]

---

## Short Comment (SC1) · 8 Mar 2018

This paper, with the main group of authors from Latin America, suggests a regional collaboration for mapping SOC across the region based on country-specific soil organic carbon (SOC) maps. This bottom-up approach will be an ideal scenario to achieve the goals of global/regional projects like GlobalSoilMap. However, the paper presented another view, a top-down approach based on the global WoSIS database. We can only speak of the case of Chile, which actually doesn't appear in the list of WoSIS collaborators: (http://www.isric.org/explore/wosis/wosis-cooperating-institutions-and-experts). In a true case of collaboration, Chile wouldn't only have 44 point observations (Table 1

of the manuscript) but more than 400 (Padarian et al., 2016). We assume that these are the same institutions that participated on FAO the Global SOC Map, which was delivered on December 2017, so the Chilean data has already being processed. Another example in Brazil, where Samuel Rosa et al. (2017) are building a collaborative nation-wide soil database from bottom-up. Such spirit of collaboration should be preferred in this era of open data.

Nevertheless, the paper attempted to compare country-specific estimates of SOC. Facilitating reproducibility is always appreciated but we don't think it justifies the use of a very coarse 5x5 km resolution, considering that current DSM studies can produce much finer resolution ranging from 100 to 1000 m for that extent.

The paper also deals with an interesting topic trying to find the right covariates for DSM, but the purpose is defeated when this study used a brute force approach of trying all 118 covariates. For example, why does mean night-time temperature, and not other temperature measures has the highest correlation for the case of Chile? Of course temperature is important in SOC dynamics, but it does that justify using 6 temperature variables (out of 10 covariates) for prediction (Table 2 of the manuscript)? A more conscious selection of relevant covariates should be stressed when developing such a regional model.

Finally, while this paper encouraged positive collaboration, it missed many country-specific SOC maps, including some from the countries listed in this paper (full disclosure: We are the co-authors of one of them. The first author of this manuscript has also done interesting work at the national extent for Mexico). To be a full collaborative project, a bottom up approach that takes full advantages of existing region-specific soil maps should be encouraged. This paper shows many methods can be used to derive SOC maps, the challenge is how to combine existing information with new digital products, or how to combine seamlessly SOC maps from different countries in a true collaborative effort.

José Padarian, Budiman Minasny

References

Padarian, J., Minasny, B. and McBratney, A.B., 2017. Chile and the Chilean soil grid: a contribution to GlobalSoilMap. Geoderma Regional, 9, pp.17-28.

Samuel-Rosa et al., 2017. The Free Brazilian Repository for Open Soil Data. https://www.researchgate.net/project/The-Free-Brazilian-Repository-for-Open-Soil-Data

---

## Referee Comment (RC2) · Anonymous Referee #2 · 10 Mar 2018

Manuscript: soil-2017-40

Comments/suggestions:

The manuscript entitled "No Silver Bullet for Digital Soil Mapping: Country-specific Soil Organic Carbon Estimates across Latin America" could be a valuable contribution to the Global SOC Mapping initiative of the UN-FAO, and probably was a result of a collaborative effort among researchers across Latin America. The objectives of the study are clear, and the methods applied are fairly common in digital soil mapping community.

I suggest the authors should consider addressing the following concerns:

[Figure]

1) It would be interesting to see a map showing the geographical distribution of SOC observations in each country. 2) Explain the country-specific pedo-transfer functions that the authors' have developed to address missing bulk density values. 3) Compare and contrast your results with other national/global SOC mapping studies. What added values did your study provide? 4) Include in table the total SOC stocks for each country you've mapped and compare the values with other published SOC estimates. 5) The authors highlighted it as a country-specific effort in mapping SOC in Latin America but when it comes to SOC observations, they only relied on WoSIS database. To me, it is more like top-down rather than a bottom-up approach, as far as the use of national SOC data is concerned. I assume some of the Latin American countries, for example, Chile (Padarian et al., 2017) holds more SOC observations than you've used in your study. Please explain.

Reference

Padarian, J., Minasny, B. and McBratney, A.B. 2017. Chile and the Chilean soil grid: a contribution to GlobalSoilMap. Geoderma Regional 9:17-28.

---

## Author Comment (AC1) · 10 Apr 2018

Responses to referee 1

The authors appreciate the overall support and constructive feedback provided by the reviewer, which will improve the clarity of the paper. We recognize that improvement is needed for the methodological sections. Please note that our conclusion about no best method has been recognized in other studies (Ho & Pepyne, 2002; Qiao et al., 2015) but it is still an ongoing discussion. We argue that there are no best methods for statistical modeling in a quantifiable basis, and we argue that the no-free lunch statistical theorem should apply also for digital soil mapping.

[Figure]

Specific response to reviewer

1. Weighted overall measure of mapping accuracy maybe more informative than stats per country?

We have calculated an overall fit using each one of the methods for all Latin America and the overall fit is ~0.41 of explained variance. That said, our overall objective was to evaluate the performance of each algorithm for each country regardless of size and information (i.e., data points). We recognize that with time the WoSIS database will grow as there are multiple sources of information within countries. Our framework assumes that we cannot conclude that RK and RF are the "best methods" if we cannot prove that in fact they are always the "best methods" for all countries (i.e., small and large countries). Although RF and RK were the 'best methods' in terms of absolute accuracy (see Taylor diagram in Figure 2 that for Mexico and Brazil (as in most of the countries), all methods (points) overlap each other and fall in the same range of error, variance and correlation. We recognize that the methods section can improve the explanation of the methods used for model evaluation and how to interpret them (e.g., Talyor diagrams, see (Taylor, 2001; Carslaw & Ropkins, 2012). Please note that an overall measure of mapping accuracy for the 5 methods across all the datasets was performed for bulk density, coarse fragments and SOC (density and mass). Figure 1 shows that all models have relatively similar prediction capacity (except for bulk density where RK generate the worse predictions), supporting the "no free lunch" theorem. We will work in a revised version to improve clarity in the methods and discussion section to make this argument clear.

2. Building models with <30 training points can lead to artifacts

We agree that this recommendation is common for geostatistics. That said, this recommendation does not necessarily apply for machine learning. Based on the quantile theory (e.g., Meinshausen, 2006) and examples for regression and classification (e.g., Zhu, 1997; Pearson et al., 2007) we argue that having just a few points (i.e., ~10 ) it is

possible to achieve reliable predictions based on a machine learning framework. We recognize that with time the WoSIS database will grow as there are multiple sources of information within countries. Consequently, our approach can be replicated and tested in the future. We strongly believe that this information represents an important baseline for Latin American countries that have limited information and training for digital soil mapping. Thus, our efforts are empowering capacity building across Latin America.

3. Is local better than global? Is it justified to stitch maps produced by countries vs using global models? (note that reviewer skipped #3 so number in answers are shifted from the reviewer comments).

We fully agree that more information and discussion about global/Latin America vs country-specific analyzes is an ongoing discussion for digital soil mapping. We ran several models for all Central America (i.e., "global" approach) and found very low accuracy and high bias ($\sim$2% of explained variance), a "global" Latin American fit improved explained variance up to $\sim$41%. We can include these results and discussion in the revised version. Please note that Figure 1 shows the comparison of the five methods applied to SOC related properties considering all available data for Latin America (i.e., "global" approach). We will make this clearer in the methods section and discussion in a revised version as suggested by the reviewer.

4. Evaluation of the accuracy of predictions should be based on e.g. 5-fold CV with re-fitting

We appreciate the comment from the reviewer and we will improve clarity of the methods in a revised version. We argue that a fair comparison (either five- or ten-fold) means that the same method was applied to all datasets. Consequently, we selected 10-fold based on previous studies (Borra & Di Ciaccio, 2010). We clarify that the model evaluation is based on interpretation of Taylor Diagrams, which are common practice for model comparison and evaluation for climate research. We will make this clearer in a revised version. We have seen that the lower density of points the lower variance

is achieved by increasing the number of folds. Previous studies have shown that under specific cases a 10-fold validation slightly decreases the variance (Markatou et al., 2005). However, a visual inspection and descriptive statistics of the training dataset is always recommended as good practice (Esbensen & Geladi, 2010) but it is not practical with very large datasets. We argue that the differences in accuracy associated with the number of folds are significantly lower than the differences between models (which use the data under different assumptions); thus, we believe that our approach is robust.

5. Github repository does not contain all points and grids you have downloaded from ISRIC WoSIS and worldgrids.org.

Our code is designed to process the original WoSIS dataset, thus we do not provide a "dataset" as the information is available in WoSIS.

Other detailed comments in the manuscript and questions are available here.

The authors will properly address the detailed comments. Note from the detailed comments that there is still large uncertainty of >5x5 km pixels soil carbon estimates (Tifati et al. 2017), which is the main reason why multi-scale mapping exercises including coarse grained maps are continuously required to better understand the sources, the spatial controls and error propagation pathways of carbon cycle-related models. The authors appreciate your constructive review. The authors believe that properly addressing your concerns along the document will benefit the value of the paper.

References

Borra S, Di Ciaccio A (2010) Measuring the prediction error. A comparison of cross-validation, bootstrap and covariance penalty methods. Computational Statistics & Data Analysis, 54, 2976–2989.

Carslaw DC, Ropkins K (2012) openair — An R package for air quality data analysis. Environmental Modelling & Software, 27–28, 52–61.

Esbensen KH, Geladi P (2010) Principles of Proper Validation: use and abuse of re-sampling for validation. Journal of Chemometrics, 24, 168–187.

Ho YC, Pepyne DL (2002) Simple Explanation of the No-Free-Lunch Theorem and Its Implications. Journal of Optimization Theory and Applications, 115, 549–570.

Markatou M, Tian H, Biswas S, Hripcsak G (2005) Analysis of Variance of Cross-Validation Estimators of the Generalization Error. Journal of Machine Learning Research, 6, 1127–1168.

Meinshausen N (2006) Quantile Regression Forests. J. Mach. Learn. Res., 7, 983–999.

Pearson RG, Raxworthy CJ, Nakamura M, Townsend Peterson A (2007) ORIGINAL ARTICLE: Predicting species distributions from small numbers of occurrence records: a test case using cryptic geckos in Madagascar. Journal of Biogeography, 34, 102–117.

Qiao H, Soberón J, Peterson AT (2015) No silver bullets in correlative ecological niche modelling: insights from testing among many potential algorithms for niche estimation (ed Kriticos D). Methods in Ecology and Evolution, 6, 1126–1136.

Taylor KE (2001) Summarizing multiple aspects of model performance in a single diagram. Journal of Geophysical Research: Atmospheres, 106, 7183–7192.

Tifafi M, Guenet B, Hatté C Large Differences in Global and Regional Total Soil Carbon Stock Estimates Based on SoilGrids, HWSD, and NCSCD: Intercomparison and Evaluation Based on Field Data From USA, England, Wales, and France. Global Biogeochemical Cycles, 2017GB005678.

Zhu A-X (1997) A similarity model for representing soil spatial information. Geoderma, 77, 217–242.

Please also note the supplement to this comment:

https://www.soil-discuss.net/soil-2017-40/soil-2017-40-AC1-supplement.pdf

[Figure]

**Supplement:**

3   Detailed response to comments from referee 1

6   **Page   Comment**

1   1      we would like to maintain the tittle as our model evaluation includes soil carbon and other related properties (see Figure 1).

9   2   1      ok

2   2      ok

2   3      model variance (as surrogate of model uncertainty) at coarse scales should not be
12   avoided just by the availability of higher resolution data, it should be rather explained and reduced (increase model agreement) with more and more validation experiments.

2   4      data scenario is referred to available data per country

15   2   5      because this countries have both rich and low SOC stocks from arid environments to highly productive ecosystems.

2   6      yes, we will improve the explanation of this step

18   2   7      there is lower accuracy on a continental model (data not shown) using all data contained in WoSIS, we will include more information on the revised version of the manuscript.

3   1      we would argue that is not a fair comparison (country vs global) but we will include
21   some other performance metrics

4   1      ok

4   2      ok

24   4   3      ok

4   4      the use of ISRIC data does not affect our conclusion of best methods and soil-related institutions in the countries currently are leading research based on country-specifics data and products.

27   4   5      ok, in supplement

4   6      ok we appreciate your contributions

30  5    2    the cross validation results were not sensitive to repeated cross validations (1 time) and between 10 or 5 fold (results varied <2%).

5    3    ok

33  5    4    yes models tend to do that and we show how using realistic and non realistic model limits can affect largely our predicted stocks

5    5    ok

36  5    6    we will change this limit number. The intention was to show how a potential misuse of prediction limits can affect our predicted stocks but we will reduce this number to something realistic.

5    7    ok

5    8    yes the weights were the ratios between rmse and r2

39  5    9    similar to r2 but different, this ratio is proposed in this work as a complementary metric for model evaluation that can be used to weight the average of different predictions.

42  6    1    we agree but the variance of different assumptions analyzing the same problem and generating multiple plausible predictions is also uncertainty. The uncertainty that you mention is just across the places we know, where we have observational data.

6    2    ok

45  6    3    ok

7    1    ok

8    1    RF variable importance was not considered here

48  9    1    ok

9    2    these are validations, we will better explain this step

9    3    the direction and magnitude of the correlation analysis

51  10   1    ok

10   2    these are rich data countries

54  10   3    carbon density is affected by the presence of both low carbon and high carbon dominated areas in large countries

10   4    to show an obvious relationship support the reliability of our framework which is meant to support capacity building across Latin America.

57  11   1    ok

Detailed response to comments from referee 1 https://doi.org/10.5194/soil-2017-40

| | 12 | 1 | ok |
|---|---|---|---|
| | 12 | 2 | ok |
| 60 | 12 | 3 | ok we will re-organize discussion section |
| | 14 | 1 | but this adds more complexity to model performance. |
| | 14 | 2 | ok |
| 63 | 14 | 3 | at the global scale too. |

---

## Author Comment (AC2) · 10 Apr 2018

Responses to referee 2

The authors appreciate the constructive and positive feedback from the referee. The authors will follow the referee suggestions in order to improve the quality and value of the paper. Below, we also clarify some concerns about our decision of using only publicly available data.

1) It would be interesting to see a map showing the geographical distribution of SOC on each country

We fully agree and will include a map of the observational data used for our model evaluation.

2) Explain the country-specific pedotransfer have developed to address missing bulk density values.

We appreciate this comment. We clarify that we did not developed any new pedotransfer function applied to missing bulk density values. To fill missing values, we used a simple pedotransfer function based on organic matter OM (Drew, 1973, BD = 1/(0,6268 + 0,0361 * OM). We decided to use the equation because it showed less extreme values than other available pedotransfer functions during preliminary training exercises (data not shown, see FAO, 2017 p7). Another reason is that there is not a single pedotransfer function applicable to all soil types across Latin America. The proposed equation is representative for soils with organic matter content between 0.17 to 13.5% (Drew, 1973). We assumed a value of 0 when coarse fragments were missing, which could lead to overestimations of SOC stocks. Because of these reasons we focus on model comparisons across 19 possible scenarios of data under a variety of environmental conditions, rather than reporting SOC stocks. Country-specific SOC stocks are required by the United Nations to be officially reported by the institutions of each country with the mandate to generate soil information with certain data-specifications (e.g., 1km or less). This effort is beyond the scope of this study as our intention is to provide a fully reproducible framework with no major computational requirements (i.e., conventional laptop) and in short periods of time (2-6 hours). This approach is meant to provide capacity building for digital soil mapping across Latin America.

3) Compare and contrast your results with other national/global SOC mapping studies. What added values your study provide?

We will enrich the discussion of our results in a revised version of the manuscript. We will mention SOC stocks but will focus on model performance and model performance metrics (as SOC stocks is not the main focus of the paper, but we will make

modifications in the results [see next response]). With this study we aim to provide a benchmark for model performance evaluation across Latin America using a publicly available dataset and publicly available covariates at a contextual resolution of 5x5km, in a country-specific basis.

4) Include a Table with the total SOC stocks for each country you've mapped and compare the values with other published estimates

We will include more information (e.g., a new Table) about the estimated soil organic carbon stocks from the predicted maps. Another possibility could be to modify Figure 4 to and report SOC stocks. We will also include a brief discussion based on previously reported estimates but will emphasize the importance of model accuracy within the available datasets.

5) Top-down rather than a bottom-up approach regarding the data used

The authors agree that this effort represent a top-bottom approach regarding the data used. The authors recognize that country-specific SOC estimates should be ideally based on all the best available information on each country. However, our effort to highlight in a country basis that the no-free-lunch theorem should also apply for soil carbon mapping does not seems to be compromised by the used data. ISRIC's soil profile dataset is a compilation of national soil inventories from a large number of nations which has been successfully used to generate global hypothesis about soil carbon dynamics (Sanderman, Hengl and Fiske. 2017), and we would argue that is a suitable dataset for a multi-model approach, such as the one presented here.

Recently the United Nations requested Latin American countries to build technical and institutional capacities on digital soil mapping, thus, each institution responsible on each country has dedicated efforts to rescue legacy data and assembling a national covariate space with the ultimate goal to implement an efficient spatial inference system customized to their country-specific needs (e.g., country level soil carbon reports). Large amount of information is still on a "rescue phase" (e.g., from paper to digital)
and/or being curated by a soil expert designated by each country.

From a bottom-up approach, each country has a unique digital soil mapping story to tell and we consider important to encourage the institutions of each country to lead scientific research around the use of publicly and non-publicly, yet available soil data sets across their countries. We believe that such effort would strength the formulation of public policy around soil resources management and country-specific soil conservation capabilities. However, we also argue that countries should learn from each other experiences and work together towards the development of reference frameworks for the country-to-region validation of global information sources and models. Such collaborative effort would be of high priority since large discrepancy has been reported on globally available soil carbon datasets at this contextual coarse resolution (Tifati, Guenet and Hatté. 2017). Thus, we believe that studying the effect of using top-bottom standardized datasets to train country-specific models represent a middle point that could contribute to explain the current discrepancies between country-specific and global soil organic carbon models.

The work of Padarian, Minasny and McBratney (2017) is an important reference framework, which will be cited in the following manuscript version. It shows a good proportion to available data across Chile. The use of Google earth for digital soil mapping is currently underutilized and may represent the best option to handle big datasets incredibly fast and with access to multi-source and multi-temporal environmental information sources relevant for soil mapping applications. We invite Padarian, Minasny and McBratney (2017) to consider the participation of Chilean institutions in future efforts to map soil properties and functions across this country. Current soil mapping efforts across large countries such as Chile with large gaps of information and sparse data would benefit from the interaction between the expertise of the authors at the University of Sydney and the institutions with the mandate to generate soil information across this nation (e.g., Ministry of Agriculture).

The authors appreciate your constructive feedback to this manuscript.

References

Drew, L. A. (1973). Bulk density estimation based on organic matter content of some Minnesota soils. Minnesota Forestry Research Notes, no. 243.

FAO. 2018. Soil Organic Carbon Mapping Cookbook. Y Yigini, GF Olmedo, K Viatkin, R Baritz, and RR Vargas, (Eds). 2nd edition, 2018.

Padarian, J., Minasny, B. and McBratney, A.B. 2017. Chile and the Chilean soil grid: a contribution to GlobalSoilMap. Geoderma Regional 9:17-28

Sanderman J, Hengl T, Fiske GJ (2017) Soil carbon debt of 12,000 years of human land use. Proceedings of the National Academy of Sciences, 114, 9575–9580.

Tifafi M, Guenet B, Hatté C Large Differences in Global and Regional Total Soil Carbon Stock Estimates Based on SoilGrids, HWSD, and NCSCD: Intercomparison and Evaluation Based on Field Data From USA, England, Wales, and France. Global Biogeochemical Cycles, 2017GB005678.

---

## Author Comment (AC3) · 10 Apr 2018

Responses to comments

We appreciate your comments and feedback to this effort. We agree that, regarding the data used, this is a "top-bottom" standardized approach. We are aware that there is more soil information (e.g., legacy pedon descriptions) on each country compared with what currently is contained in the WoSIS system. Most of these country-specific information has been used by each representative institution to deliver country-specific soil carbon information for the Global Soil Partnership initiative (GSP). To support these activities, the GSP has dedicated efforts to identify the institutions and individuals on

each country with the mandate to generate and update soil information with a national perspective. These institutions and individuals were the participants of a series of training sessions on digital soil mapping and this study is one result of this collective effort. A spirit of transparent methods, data-sharing and recognition of the hard work across the institutions to provide the nation-wide datasets useful for digital soil mapping are also welcome ideas to improve bottom-up digital soil mapping practices. To empower institutions with state-of-the-art approaches to handle big data such as the Google earth based framework described by Padarian et al. (2016) would be also beneficial to progress with nation-wise digital soil mapping assessments. Please note that each country is facing different challenges mapping their own soil resources based on their own country-specific needs.

We appreciate your comment about variable selection, and we fully agree. Our point is to encourage the inclusion of a variable selection strategy before model soil properties, which would benefit both, the model interpretability and for the case of machine learning, it would also simplify the computational demand. We will clarify this idea in the revised manuscript. A variable selection strategy can be as simple as a correlation analysis or as complex as a genetic search algorithm for variable selection. For the first case (which we used), we will highlight in the revised version that, ideally, the best correlated selected predictors should be chosen in a source specific basis (e.g., the best climatic, the best topographic, the best vegetation index and so on). Please consider that some machine learning algorithms (e.g., random forest) do not have the assumption of multicolinearity in the covariate space. In addition, we are performing cross-validation and therefore, obtaining unbiased residuals, which supports the performance report of our predictions.

As explained in the responses to referee 2, from a bottom-up approach each country has important progress to report. Our conclusion is not affected by the use of the WoSIS dataset. From datasets harmonization efforts to parallel computing problems each Latin American country is developing capacities for digital soil mapping, not only

[Figure]

Brazil or Chile. The ultimate goal would be to reduce the current uncertainty in the carbon cycle related estimates from country-specific-to-regional-(Latin America)-to-global scales. We agree that pioneer efforts such as yours, or Samuel-Rosa et al. (2017) for Brazil, or Angelini et al., (2017) in Argentina and others, should be cited in a revised version of our work. Please note that there are still large uncertainty on SOC across the multiple-scales of data availability (e.g., from 250 pixels to ∼5x5km, Tifafi et al. 2017), so step by step we will address finer resolution digital soil mapping, hand by hand with the responsible institutions on each country, with the ultimate goal to build capacities on digital soil mapping. We would be delighted if you are interested in to contribute with your experience on this collective and true collaborative effort.

References

Angelini M. E., Heuvelink G. B. M., Kempen B. (2017) Multivariate mapping of soil with structural equation modelling. European Journal of Soil Science, 68, 575–591.

Padarian, J., Minasny, B. and McBratney, A.B., 2017. Chile and the Chilean soil grid: a contribution to GlobalSoilMap. Geoderma Regional, 9, pp.17-28.

Samuel-Rosa et al., 2017.The Free Brazilian Repository for Open Soil Data. https://www.researchgate.net/project/The-Free-Brazilian-Repository-for-Open-Soil-Data

Tifafi M, Guenet B, Hatté C Large Differences in Global and Regional Total Soil Carbon Stock Estimates Based on SoilGrids, HWSD, and NCSCD: Intercomparison and Evaluation Based on Field Data From USA, England, Wales, and France. Global Biogeochemical Cycles, 2017GB005678.

---

## Author Response (AR1)

We appreciate the feedback from the Topical editor and the reviewers, which will increase the value of the paper. We have addressed all comments and concerns from the reviewers and Topical editor. Major changes include the fit of a regional (Latin American) model and the comparison (with global estimates) of SOC stocks at the contextual resolution of 5x5km grids. We compare the resulting SOC stocks from our modeling approach in a country-specific basis with regional SOC models for Latin America and with previously reported SOC from global estimates (GSOCmapGSP, SoilGrids, and HWSD). We have also included a simple linear combination of the different modeling approaches by stacking the different predictions using a linear blend of models that we used to maximize the prediction accuracy. Our results suggest that global estimates predict higher SOC stocks than country-specific maps. We provide a reproducible example for SOC mapping that is currently being used for building capacities for digital soil mapping across Latin America with the ultimate goal of reducing uncertainties in the global caron cycle.

---

## Author Response (AR2)

**Response to Topical Editor**
**Manuscript soil-2017-40**

*No Silver Bullet on Digital Soil Mapping: Country-specific Soil Organic Carbon Estimates across Latin America*

*The authors acknowledge the revision and comments from the Topical Editor. We have revised English errors and improved the overall quality of the paper, thanks to the valuable feedback previously received.*

*Guevata et al., 2018.*